EMBO
Molecular Medicine

# CoQ deficiency causes disruption of mitochondrial sulfide oxidation, a new pathomechanism associated with this syndrome

Marta Luna-Sánchez[1,2,*,†], Agustín Hidalgo-Gutiérrez[1,2,†], Tatjana M Hildebrandt[3], Julio Chaves-Serrano[2], Eliana Barriocanal-Casado[1,2], Ángela Santos-Fandila[4], Miguel Romero[5], Ramy KA Sayed[2,6], Juan Duarte[5], Holger Prokisch[7], Markus Schuelke[8], Felix Distelmaier[9], Germaine Escames[1,2], Darío Acuña-Castroviejo[1,2] & Luis C López[1,2,**]

## Abstract

Coenzyme Q (CoQ) is a key component of the mitochondrial respiratory chain, but it also has several other functions in the cellular metabolism. One of them is to function as an electron carrier in the reaction catalyzed by sulfide:quinone oxidoreductase (SQR), which catalyzes the first reaction in the hydrogen sulfide oxidation pathway. Therefore, SQR may be affected by CoQ deficiency. Using human skin fibroblasts and two mouse models with primary CoQ deficiency, we demonstrate that severe CoQ deficiency causes a reduction in SQR levels and activity, which leads to an alteration of mitochondrial sulfide metabolism. In cerebrum of *Coq9[R239X]* mice, the deficit in SQR induces an increase in thiosulfate sulfurtransferase and sulfite oxidase, as well as modifications in the levels of thiols. As a result, biosynthetic pathways of glutamate, serotonin, and catecholamines were altered in the cerebrum, and the blood pressure was reduced. Therefore, this study reveals the reduction in SQR activity as one of the pathomechanisms associated with CoQ deficiency syndrome.

**Keywords** blood pressure; COX; glutathione; mitochondrial disease; SQR
**Subject Categories** Genetics, Gene Therapy & Genetic Disease; Metabolism

See also: **M Ziosi** *et al* (January 2017)

## Introduction

Primary and secondary coenzyme Q (CoQ) deficiencies are clinically and genetically heterogeneous, with muscle, kidneys, and central nervous system being the main affected organs and systems. Because of the function of CoQ as antioxidant and as electron carrier in the mitochondrial respiratory chain, increases in oxidative stress and/or a decline in mitochondrial energy production and pyrimidine biosynthesis have been identified as pathomechanisms of the disease (Quinzii *et al*, 2008, 2010; Emmanuele *et al*, 2012).

Besides the antioxidant and bioenergetics functions, CoQ links the mitochondrial electron transport chain to the TCA cycle by succinate dehydrogenase (EC 1.3.5.1), to β-oxidation by electron-transfer flavoprotein:ubiquinone oxidoreductase (EC 1.5.5.1), to the shuttling of reduction equivalents from the cytoplasm by glycerol-3-phosphate dehydrogenase (EC 1.1.99.5), to the synthesis of pyrimidines by dihydroorotate dehydrogenase (EC 1.3.3.1), to the metabolism of glycine by choline dehydrogenase (EC 1.1.99.1), to arginine and proline metabolism by proline dehydrogenase (EC 1.5.99.8), and to sulfide metabolism by sulfide:quinone oxidoreductase (EC 1.8.99.1; SQR). However, the relative contributions of these pathways to the overall pathophysiology of CoQ deficiency have not been evaluated so far, and only a defect in pyrimidine biosynthesis in cell culture and a decline in the steady-state levels of SQR protein in a proteomic analysis on heart and kidney of *Coq9[R239X]* mice have been reported (Lopez-Martin *et al*, 2007; Lohman *et al*, 2014).

Sulfide:quinone oxidoreductase catalyzes the first step in the mitochondrial sulfide oxidation pathway. In this reaction, $H_2S$ is oxidized

1  Departmento de Fisiología, Facultad de Medicina, Universidad de Granada, Granada, Spain
2  Instituto de Biotecnología, Centro de Investigación Biomédica, Universidad de Granada, Granada, Spain
3  Institut für Pflanzengenetik, Leibniz Universität Hannover, Hannover, Germany
4  Abbott Nutrition, R&D, Abbott Laboratories, Granada, Spain
5  Departmento de Farmacología, Facultad de Farmacia, Instituto de Investigación Biosanitaria de Granada, Universidad de Granada, Granada, Spain
6  Department of Anatomy and Embryology, Faculty of Veterinary Medicine, Sohag University, Sohag, Egypt
7  Institute of Human Genetics, Technische Universität München, München, Germany
8  Department of Neuropediatrics, Charité-Universitätsmedizin Berlin, Berlin, Germany
9  Department of General Pediatrics, Heinrich-Heine-University, Düsseldorf, Germany
*Corresponding author. Tel: +34 958241000 ext 20197; E-mail: martalunasan@ugr.es
**Corresponding author. Tel: +34 958241000 ext 20197; E-mail: luisca@ugr.es
†These authors contributed equally to this work

 

by SQR, forming a protein-bound persulfide. Two electrons from the oxidation of $H_2S$ are transferred via flavin adenine dinucleotide to CoQ and then to the electron transport chain. The SQR-bound persulfide is transferred to an acceptor such as glutathione (GSH) or sulfite, resulting in the generation of GSH persulfide (GSSH) or thiosulfate, respectively (Fig 1). The persulfide group from GSSH is oxidized to sulfite by a sulfur dioxygenase (EC 1.13.11.18; SDO) (also known as ETHE1 or persulfide dioxygenase). Sulfite can then either be oxidized to sulfate by sulfite oxidase (EC 1.8.3.1; SO) or converted to thiosulfate via addition of a persulfide catalyzed by the thiosulfate sulfurtransferase or rhodanese (EC 2.8.1.1; TST). The sulfane sulfur from thiosulfate can be remobilized by another sulfurtransferase called thiosulfate reductase (EC 2.8.1.3; TR) (Hildebrandt & Grieshaber, 2008; Kabil *et al*, 2014; Libiad *et al*, 2014; Di Meo *et al*, 2015).

How low levels of CoQ affect SQR activity has been studied in *Schizosaccharomyces pombe* mutant strains Δdps1 (homologue to *PDSS1*) and Δppt1 (homologue to *COQ2*). In those mutant strains, high accumulation of sulfide was reported (Uchida *et al*, 2000; Zhang *et al*, 2008). In fission yeasts, sulfide is required for the biosynthesis of both methionine and cysteine (Brzywczy *et al*, 2002). This latter aminoacid is required to synthesize glutathione, an important antioxidant in mammalian cells. Curiously, CoQ-deficient fission yeasts require cysteine and glutathione to grow on minimal medium (Uchida *et al*, 2000).

In mammals, hydrogen sulfide is increasingly being recognized as an important signaling molecule in both nervous and cardiovascular

systems. Increase in hydrogen sulfide levels induces an increase in the concentration of serotonin and a decrease in norepinephrine, aspartate, GABA, and glutamate (Skrajny *et al*, 1992; Roth *et al*, 1995). In the cardiovascular system, hydrogen sulfide induces smooth muscle relaxation and enhances vasodilation (Kabil *et al*, 2014). Moreover, accumulation of hydrogen sulfide due to mutations in *ETHE1* has been associated with cytochrome oxidase (COX) deficiency in ethylmalonic encephalopathy (Tiranti *et al*, 2009).

Based on those data, CoQ deficiency could induce a decrease in SQR activity with a concomitant increase in hydrogen sulfide levels, a fact that may contribute to the pathophysiology of CoQ deficiency. Thus, in this study, we evaluate mitochondrial hydrogen sulfide metabolism in cell and mouse models of CoQ deficiency due to mutations in different CoQ biosynthetic genes (Appendix Fig S1), with the aim of elucidating the pathophysiological consequences of an alteration in this pathway.

## Results

### Primary CoQ deficiency causes a decline in SQR activity

As previously reported, the two mouse models of CoQ deficiency used in this study, $Coq9^{R239X}$ and $Coq9^{Q95X}$, have different residual CoQ levels, resulting in different clinical phenotypes (Luna-Sanchez *et al*, 2015). While $Coq9^{R239X}$ mice have 10–15% of residual CoQ

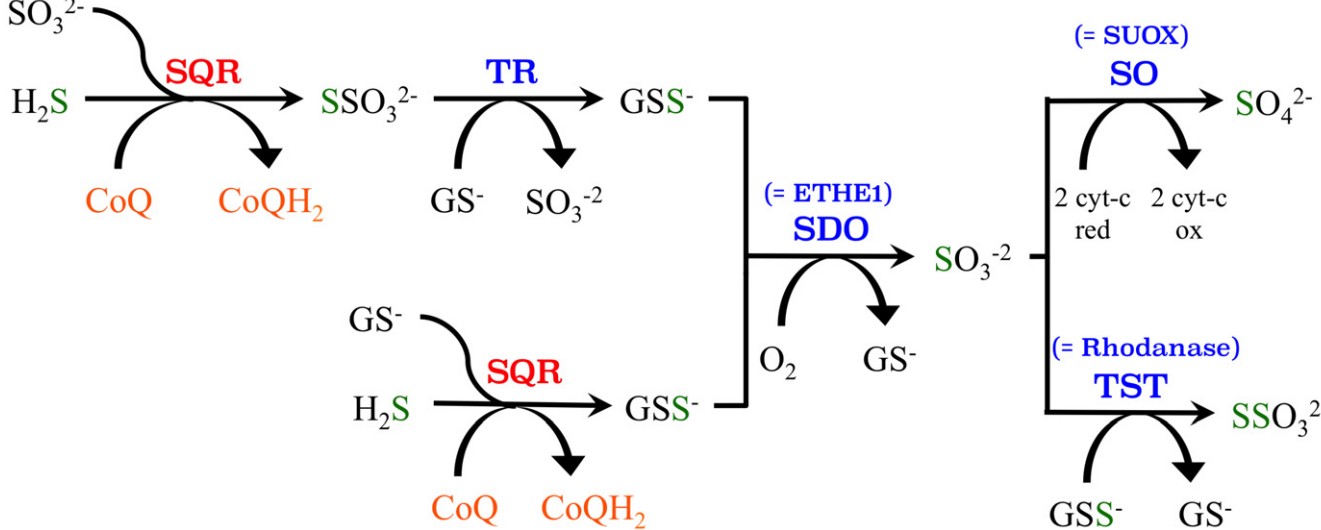

**Figure 1. Hydrogen sulfide oxidation pathway in mammalian mitochondria.**
SQR, sulfide:quinone oxidoreductase; TR, thiosulfate reductase; SDO, sulfur deoxygenase or ETHE1; SO, sulfite oxidase or SUOX; TST, thiosulfate sulfurtransferase or rhodanase.

levels in cerebrum and kidney, $Coq9^{Q95}$ mice have 40–50% of residual CoQ in the same tissues. In muscle, however, both mouse models show similar CoQ levels, 10–20% as compared to wild-type animals (Luna-Sanchez et al, 2015). As a result, $Coq9^{R239X}$ mice develop a fatal mitochondrial encephalopathy, while $Coq9^{Q95X}$ mice develop a late-onset mild mitochondrial myopathy (Garcia-Corzo et al, 2013; Luna-Sanchez et al, 2015). Because CoQ acts as an electron acceptor in the reaction catalyzed by SQR in the mitochondrial sulfide oxidation pathway, we first evaluated whether the deficit in CoQ could affect mitochondrial $H_2S$ metabolism in these mouse models. Thus, we first checked the consequences of low CoQ levels on cerebral, renal, and muscular SQR of $Coq9^{R239X}$ and $Coq9^{Q95X}$ mice at 3 months of age. The levels of $Sqr$ mRNA were similar between the three experimental groups (Fig 1A–C), and only a reduction (by 25%) in kidneys of $Coq9^{R239X}$ mice was detected (Fig 1B). At the protein level, the changes were more dramatic. The levels of SQR protein were significantly reduced in cerebrum, kidneys, and muscle of $Coq9^{R239X}$ mice, while $Coq9^{Q95X}$ mice only showed reductions in the levels of SQR in muscle (Fig 1D and F). In parallel to the reduction in SQR protein levels, the activity of SQR was significantly reduced in kidneys and muscle of $Coq9^{R239X}$ and $Coq9^{Q95X}$ mice (Fig 1G and H). While in kidneys, SQR activity in $Coq9^{R239X}$ mice was lower than $Coq9^{Q95X}$ mice (Fig 1E), SQR activity was similar in muscle of both mutant mouse strains (Fig 1F). In the cerebrum, the SQR protein was only detected in isolated mitochondria (Fig 1D) and no band was detected in tissue homogenate. This reflects the low abundance of SQR in brain (Geiger et al, 2013) (http://pax-db.org/protein/2093754/Sqrdl). Accordingly, we were not able to measure SQR activity in cerebrum.

Based on the results obtained in $Coq9$ mutant mice, where SQR levels and function were decreased in correlation with the residual CoQ levels, we next evaluated if this alteration was also present in primary CoQ-deficient skin fibroblasts due to mutations in different CoQ biosynthetic genes. We used skin fibroblasts of patients with severe CoQ deficiency due to mutation in $PDSS2$, $COQ2$, $COQ4$, and $COQ9$ (Appendix Table S1 and Appendix Fig S2A). The four mutant cells showed a reduction in the levels of SQR protein compared to control cells (Fig 2B). While the treatment with 5 μM of $CoQ_{10}$ for 1 day did not increase the SQR levels, after 7 days of treatment the levels of SQR increased in the four mutant cells, confirming that the low levels of $CoQ_{10}$ were responsible for the SQR deficiency (Fig 2B). These changes in SQR levels over the time after $CoQ_{10}$ supplementation correlate with the increase in ATP levels previously reported (Lopez et al, 2010). In vivo, supplementation with ubiquinol-10 (240 mg/kg bw/day) during 2 months in the $Coq9^{R239X}$ mouse model increased the SQR levels in muscle (Fig 3F), while in kidneys, a trend toward increase was observed (Fig 3E). These changes correlate with the increase in CoQ levels on those tissues after ubiquinol-10 supplementation (Fig 3C and D).

### Low SQR activity induces changes in the proteins involved in the mitochondrial hydrogen sulfide oxidation pathway

To know whether SQR deficiency has some impact on the mitochondrial hydrogen sulfide oxidation pathway, we first evaluated the levels and activity of TST, which also takes part in this pathway, in the tissues of mutant mice. An increase in TST levels was detected in cerebrum of $Coq9^{R239X}$ mice (Fig 4A), while in kidneys and

muscle, the differences were not statistically significant (Fig 4B and C). Nevertheless, TST activity was significantly increased in both cerebrum and kidneys of $Coq9^{R239X}$ mice, while TST activity in the same tissues of $Coq9^{Q95X}$ mice was similar to the activity in control mice (Fig 4D and E). In muscle, where the activity of TST is lower than in other two tissues, the TST activity was not altered in both mutant mouse strains (Fig 4F). $Coq9^{+/+}$, $Coq9^{R239X}$, and $Coq9^{Q95X}$ mice showed similarities in the levels of ETHE1 (SDO) in cerebrum, kidneys, and muscle (Fig 4G–I). The levels of SUOX (SO) were significantly higher in cerebrum of $Coq9^{R239X}$ mice compared to those levels in $Coq9^{+/+}$ and $Coq9^{Q95X}$ mice (Fig 4J). In kidneys and muscle, however, the levels of SUOX were similar in the three experimental groups (Fig 4K and L).

### Disruption of the mitochondrial sulfide oxidation pathway induces changes in thiols levels and metabolic disturbances in the cerebrum of $Coq9^{R239X}$ mice

Because an alteration in the metabolization of hydrogen sulfide may affect the levels of thiols, we then measured the levels of sulfides, thiosulfate, sulfite, and glutathione in mouse tissues. The levels of sulfides in the cerebrum of mutant mice were similar to those levels in control mice (Fig 5A and B). In kidneys, however, the levels of sulfides in $Coq9^{R239X}$ mice were higher than in $Coq9^{+/+}$ and $Coq9^{Q95X}$ mice (Fig 5A and B). The levels of thiosulfate and sulfite were below the detection limit (5 μM) in all tested tissues.

The major non-protein thiol in cells is GSH, which is synthesized in the cytosol and imported into mitochondria and into other organelles, where it plays an essential role in the antioxidant defense against reactive oxygen species (ROS) (Ribas et al, 2014). The levels of total GSH in the cytosol of cerebrum, muscle, and kidney were significantly decreased in $Coq9^{R239X}$ mice (Fig 6A and Appendix Fig S4A and B). However, the total GSH levels in mitochondria were normal in the three tissues of $Coq9^{R239X}$ mice (Fig 6A and Appendix Fig S2A and B). The activities of the GSH-utilizing enzymes, GPx and GRd, were significantly decreased in the cytosolic fraction of cerebrum of $Coq9^{R239X}$ mice compared to wild-type animals (Fig 6B), while in muscle and kidney, the differences were statistically significant for the renal GPx (Appendix Fig S2C and D). The decrease in GPx and GRd activities in cerebrum of $Coq9^{R239X}$ was due to a decline in the levels of GPx4 and GRd (Fig 6C and D). These decreases in the glutathione enzymes did not modify the GSSG/GSH ratios in cytosol and mitochondria (Appendix Fig S3). Therefore, our results show a global depletion in the glutathione system in cerebrum of $Coq9^{R239X}$, a mouse model of mitochondrial encephalopathy with severe histopathological signs of spongiform degeneration and reactive astrogliosis in the cerebrum.

The GSH depletion in the cerebrum may be due to a decrease in the levels of glutamate, one of the three amino acids components of GSH, with the parallel increase in N-acetylglutamate (Fig 6E). In addition, we found an increase in metabolites (L-tryptophan, 5-HIAA, and N-acetyltryptophan) of serotonin biosynthesis (Fig 6E) and a decrease in L-tyrosine (Fig 6E), which is essential in the biosynthesis of dopamine, norepinephrine, and epinephrine, in the cerebrum of $Coq9^{R239X}$ mice.

To know whether the SQR deficiency is generally responsible for the depletion in the glutathione system, we measured GSH after $Sqr$ silencing in Hepa1c1c7 cells. The silencing induced a significant decrease in the levels of $Sqr$ mRNA (Appendix Fig S4A) and SQR

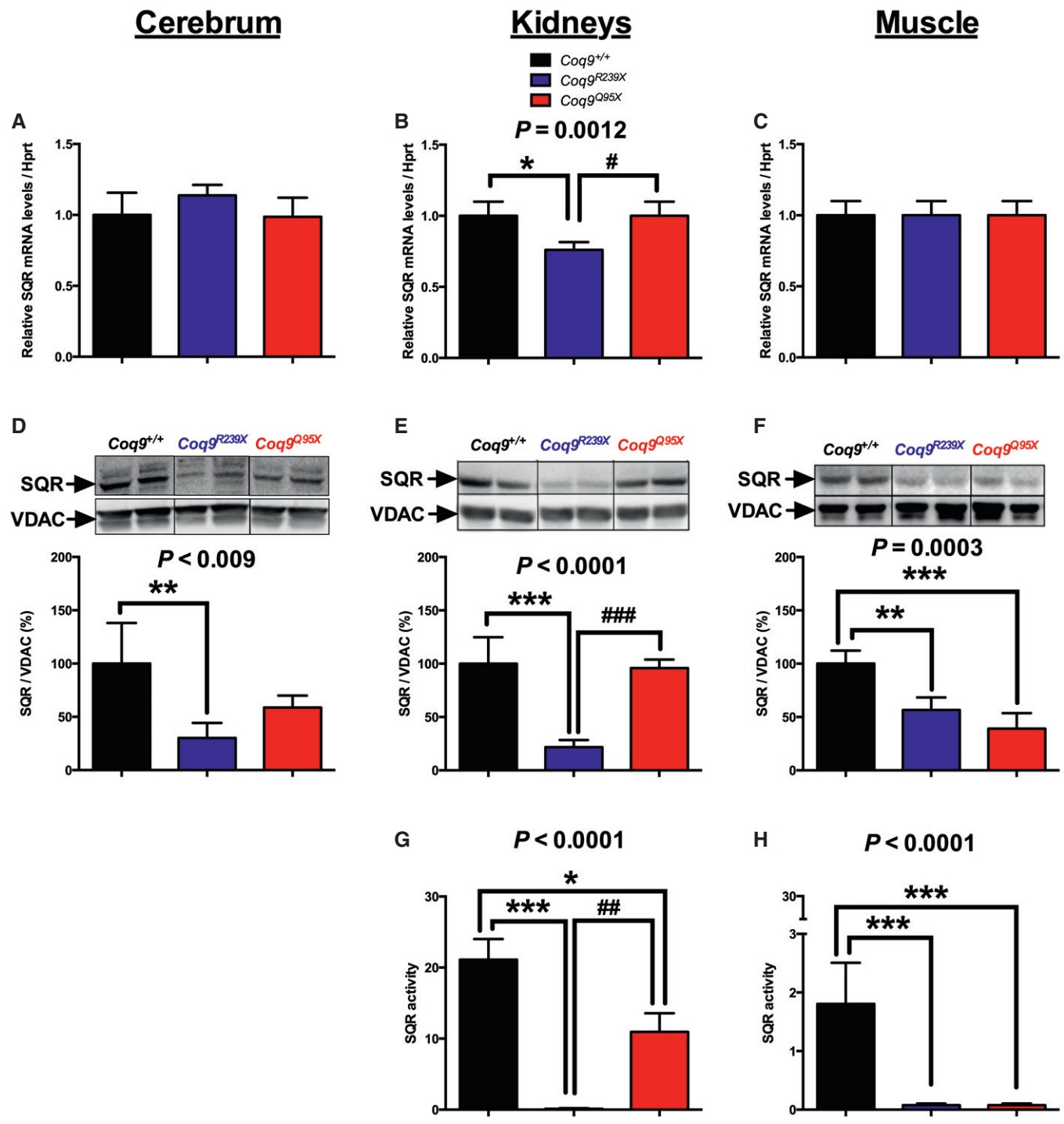

**Figure 2.  SQR levels and activity depend on CoQ levels in mice tissues.**

A–H   *Sqr* mRNA levels (A–C), SQR protein levels (D–F), and SQR activity (G, H) in cerebrum (A, D), kidneys (B, E, G), and muscle (C, F, H) of *Coq9*[+/+], *Coq9*[R239X], and *Coq9*[Q95X] mice. Note that SQR Western blots were performed in isolated cerebral mitochondria due to the low levels of this protein in cerebrum. In kidneys and muscle, the Western blots were performed in tissue homogenates. Data are expressed as mean ± SD. *$P < 0.05$; **$P < 0.01$; ***$P < 0.001$; *Coq9*[R239X] and *Coq9*[Q95X] mice versus *Coq9*[+/+] mice. #$P < 0.05$; ##$P < 0.01$; ###$P < 0.001$; *Coq9*[R239X] versus *Coq9*[Q95X] mice (one-way ANOVA with a Tukey's *post hoc* test; $n = 5$–9 for each group).

Source data are available online for this figure.

protein (Appendix Fig S4B). However, the total GSH did not change in Hepa1c1c7 cells after *Sqr* silencing (Appendix Fig S4C). In the mutant skin fibroblast with primary CoQ deficiency, where the SQR deficiency is chronic but less severe than in mice tissues or after *Sqr* silencing, the GSH levels were also similar to those levels in control fibroblasts (Appendix Fig S4D).

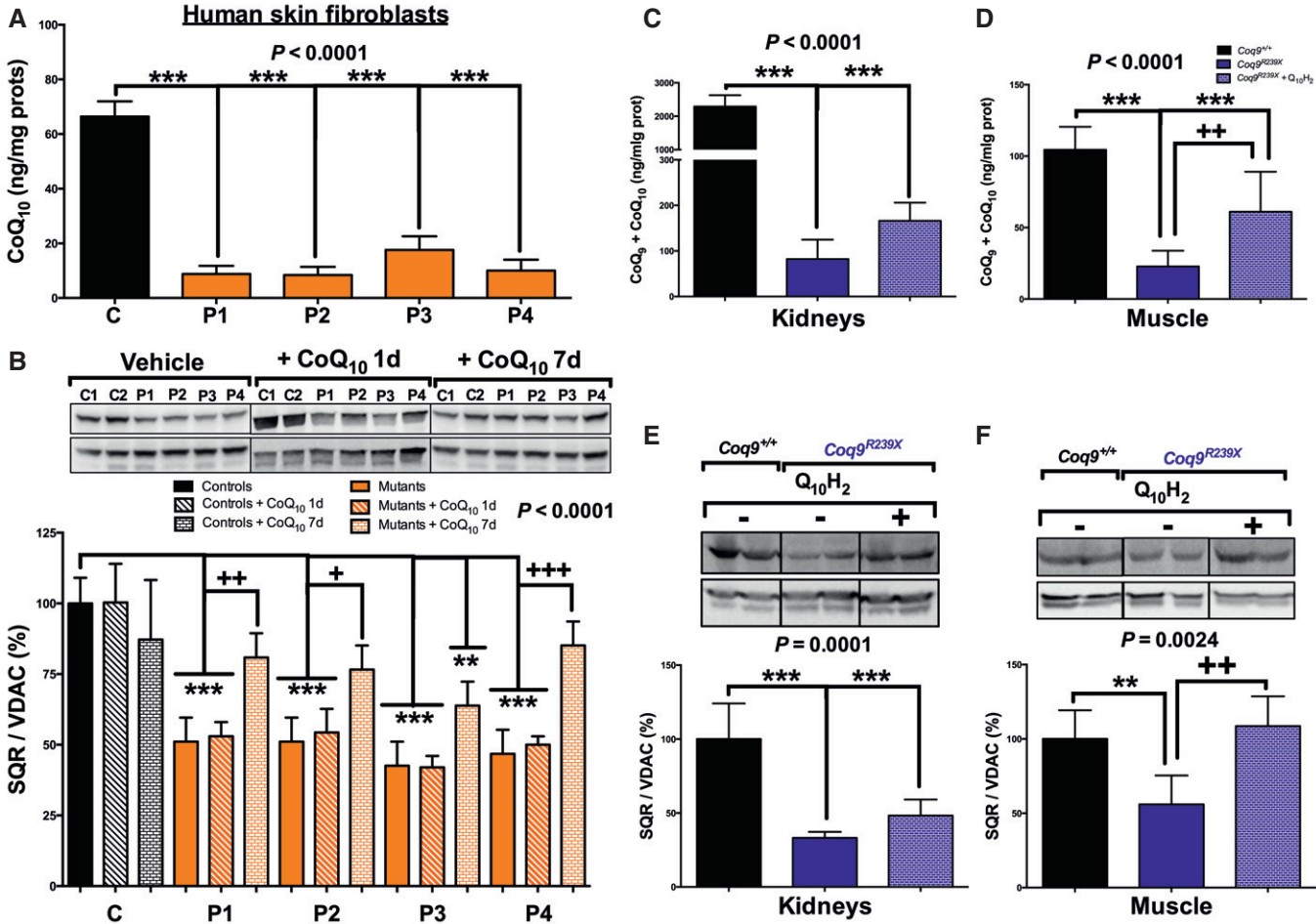

**Figure 3.** Human skin fibroblasts and mouse tissues with primary CoQ₁₀ deficiency exhibit increased SQR protein levels after exogenous CoQ₁₀ supplementation.

A    Levels of CoQ₁₀ in fibroblasts of controls (C) and patients (P1-4) with primary CoQ₁₀ deficiency.

B    Levels of SQR protein in fibroblasts of controls (C) and patients (P1-4) with primary CoQ₁₀ deficiency cultured without (vehicle) and with 5 μM of CoQ₁₀ (+ CoQ₁₀) during 1 day or 7 days.

C, D    Total CoQ levels (CoQ₉ + CoQ₁₀) in kidneys (C) and muscle (D) of $Coq9^{+/+}$, $Coq9^{R239X}$, and $Coq9^{R239X}$ + ubiquinol-10 (Q₁₀H₂) mice.

E, F    Levels of SQR protein in kidneys (E) and muscle (F) of $Coq9^{+/+}$, $Coq9^{R239X}$, and $Coq9^{R239X}$ + ubiquinol-10 (Q₁₀H₂) mice.

Data information: Data are expressed as mean ± SD. **$P < 0.01$; ***$P < 0.001$; patients versus controls, as well as $Coq9^{R239X}$ versus $Coq9^{+/+}$ mice. +$P < 0.05$; ++$P < 0.01$; +++$P < 0.001$; + CoQ₁₀ versus vehicle (one-way ANOVA with a Tukey's *post hoc* test; $n = 3$–5 for each group).

Source data are available online for this figure.

## H₂S supplementation in wild-type animals induces changes in neurotransmitters levels but does not alter mitochondrial sulfide oxidation pathway

The changes observed in the levels of the proteins involved in mitochondrial sulfide oxidation pathway, as well as in the levels of neurotransmitters in cerebrum of $Coq9^{R239X}$ mice may be attributed to the increase in hydrogen sulfide. Thus, we treated control fibroblast and wild-type mice with the H₂S donor GYY4137. After the treatment, the levels of SQR (Fig 7A) and TST (Fig 7B) did not change in control fibroblasts. Also, TST levels were similar in kidneys (Fig 7C) and cerebrum (Fig 7D) after the treatment compared with the levels in untreated animals. Nevertheless, GYY4137 induced a decrease in the levels of L-Glu and DA and an increase in the levels of

5-HIAA in cerebrum of $Coq9^{+/+}$ mice (Fig 7E), a similar tendency observed in the cerebrum of $Coq9^{R239X}$ mice (Fig 6E).

## Pathophysiological consequences of reductions in CoQ levels and SQR activity

It has been proposed that hydrogen sulfide metabolism influences COX activity and regulates blood pressure (Kabil *et al*, 2014). COX activity was, however, normal in cerebrum, kidneys, and muscle of $Coq9^{R239X}$ and $Coq9^{Q95X}$ mice (Fig 8A). COX stain did not show any signs of COX deficiency in the gastrocnemius muscle of $Coq9^{R239X}$ and $Coq9^{Q95X}$ mice (Fig 8B).

The measurement of blood pressure in $Coq9^{R239X}$ mice revealed a decrease in systolic and diastolic blood pressure

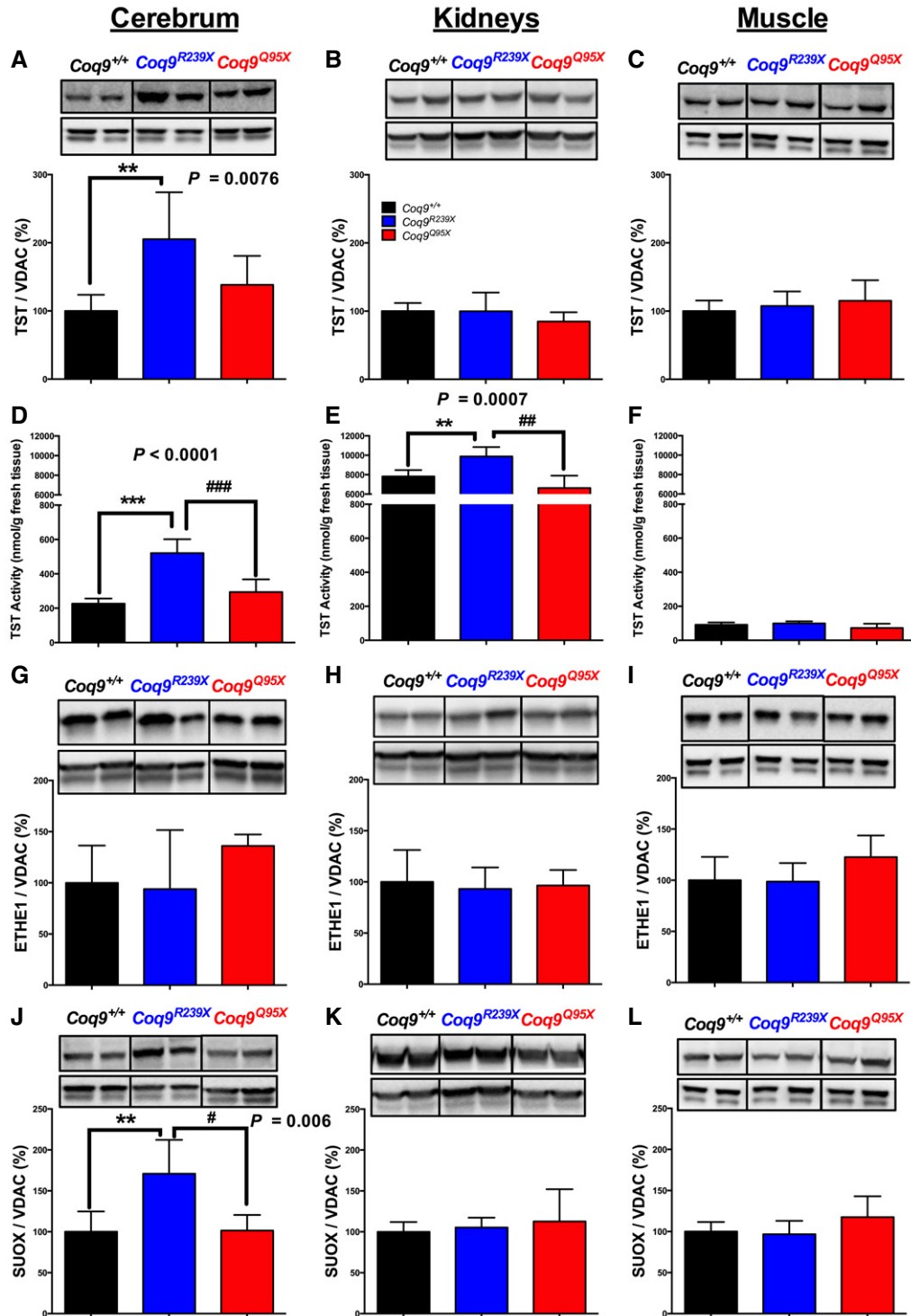

**Figure 4.   Changes in the proteins involved in the mitochondrial sulfide oxidation pathway in response to SQR deficiency in *Coq9^{R239X}* mice.**

A–C   TST protein levels in cerebrum (A), kidneys (B), and muscle (C) of *Coq9^{+/+}*, *Coq9^{R239X}*, and *Coq9^{Q95X}* mice.

D–F   TST activity in cerebrum (D), kidneys (E), and muscle (F) of *Coq9^{+/+}*, *Coq9^{R239X}*, and *Coq9^{Q95X}* mice.

G–I   ETEH1 (SDO) protein levels in cerebrum (G), kidneys (H), and muscle (I) of *Coq9^{+/+}*, *Coq9^{R239X}*, and *Coq9^{Q95X}* mice.

J–L   SUOX protein levels in cerebrum (J), kidneys (K), and muscle (L) of *Coq9^{+/+}*, *Coq9^{R239X}* and *Coq9^{Q95X}* mice.

Data information: Images in panels (C, F, and I) were obtained from the same membrane after stripping and re-blotting. Data are expressed as mean ± SD. *$P < 0.05$; **$P < 0.01$; ***$P < 0.001$; *Coq9^{R239X}* and *Coq9^{Q95X}* mice versus *Coq9^{+/+}* mice. #$P < 0.05$; ##$P < 0.01$; ###$P < 0.001$; *Coq9^{R239X}* versus *Coq9^{Q95X}* mice (one-way ANOVA with a Tukey's *post hoc* test; $n = 5–9$ for each group).

Source data are available online for this figure.

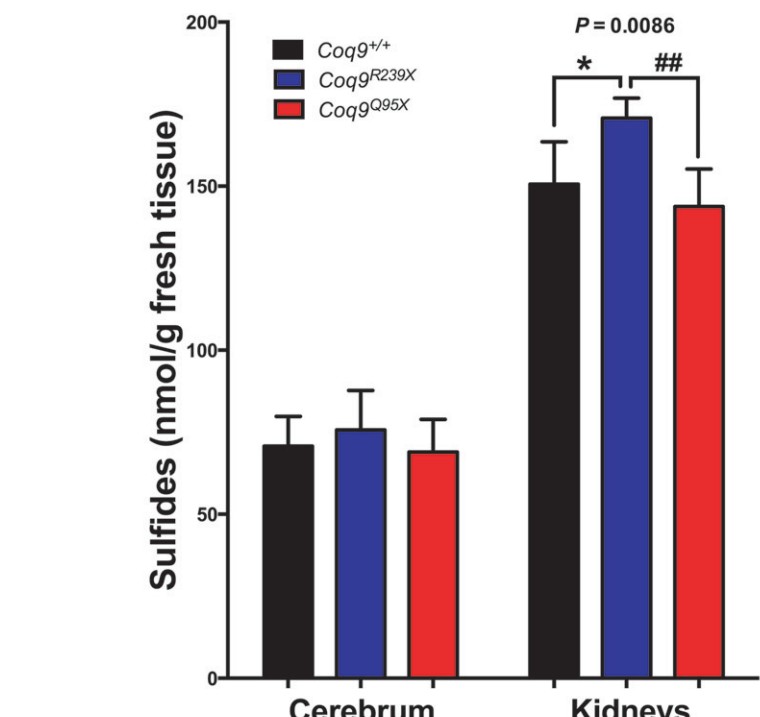

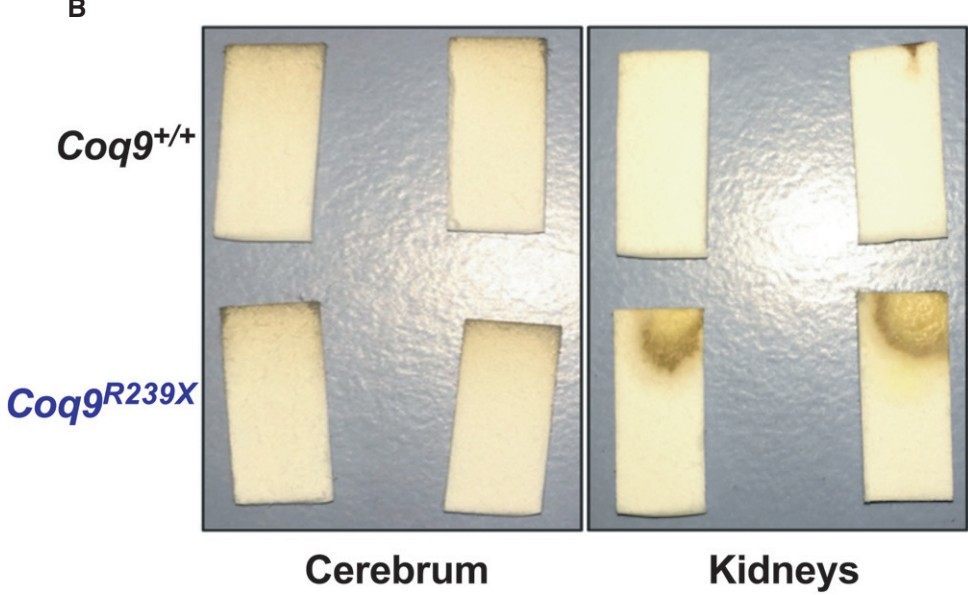

**Figure 5.  Tissue levels of sulfides in CoQ-deficient mice.**

A    Quantification of sulfide levels in cerebrum and kidneys of $Coq9^{+/+}$, $Coq9^{R239X}$, and $Coq9^{Q95X}$ mice. Data are expressed as mean ± SD. *$P$ < 0.05; $Coq9^{R239X}$ and $Coq9^{Q95X}$ mice versus $Coq9^{+/+}$ mice. ##$P$ < 0.01; $Coq9^{R239X}$ versus $Coq9^{Q95X}$ mice (one-way ANOVA with a Tukey's *post hoc* test; $n$ = 5–10 for each group).
B    Qualitative measurement of hydrogen sulfide in cerebrum and kidneys of $Coq9^{+/+}$ and $Coq9^{R239X}$ mice.

as compared to age-matched wild-type animals (Fig 9A). These changes were not related to the heart rate because the beats per minute were similar in $Coq9^{R239X}$ and $Coq9^{+/+}$ mice (Fig 9B).

## Discussion

Sulfide:quinone oxidoreductase is a mitochondrial enzyme that requires CoQ as acceptor of electrons, thereby feeding electrons

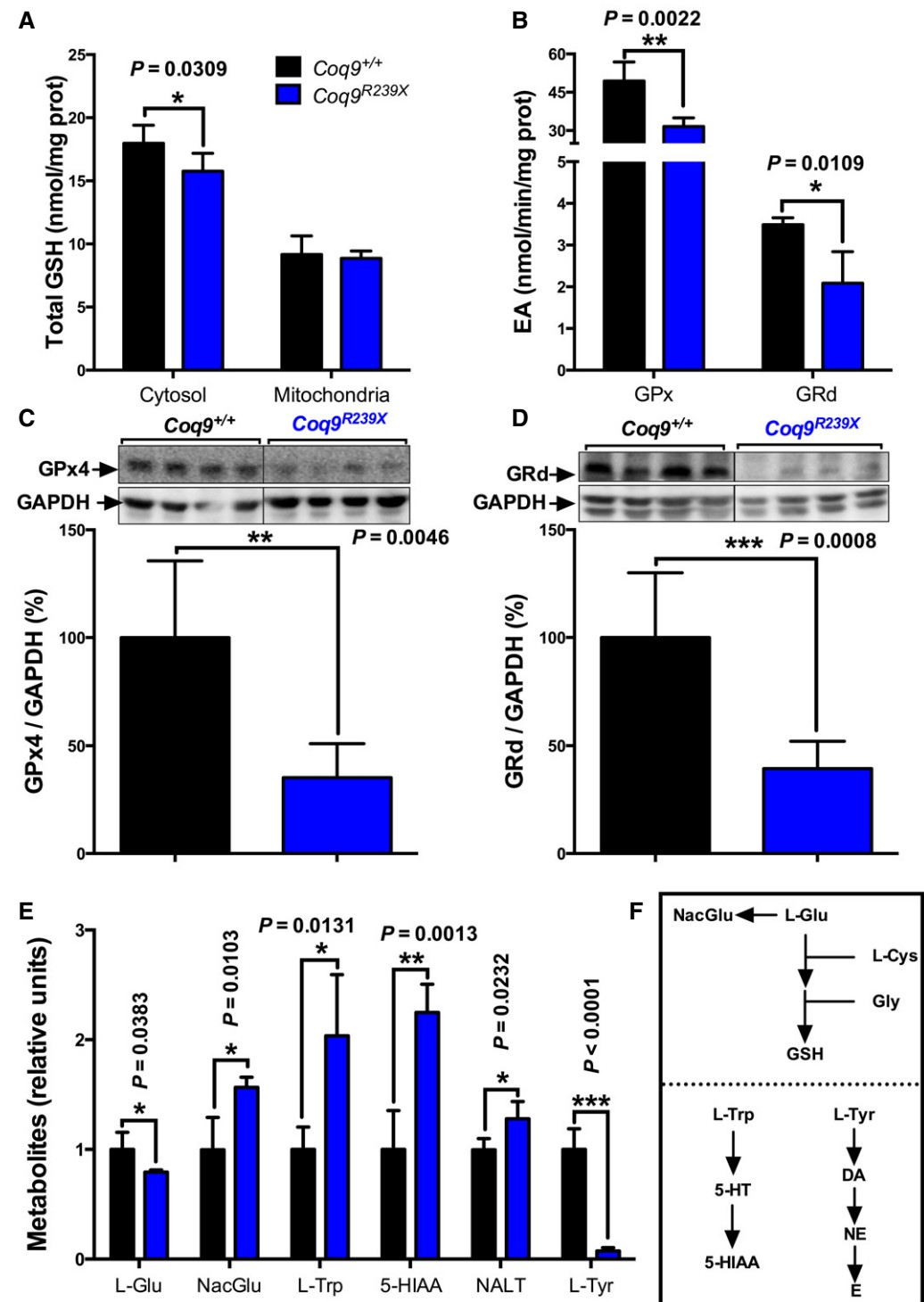

**Figure 6.  Glutathione system and neurotransmitters biosynthesis are compromised in cerebrum of *Coq9^R239X* mice.**

A    Total GSH in cytosol and mitochondria of cerebrum of *Coq9^+/+* and *Coq9^R239X* mice.
B    Cytosolic GPx and GRd activities in cerebrum of *Coq9^+/+* and *Coq9^R239X* mice.
C, D   Levels of GPx4 (C) and GRd (D) protein in cerebral homogenate of *Coq9^+/+* and *Coq9^R239X* mice.
E    Levels of L-glutamate (L-Glu), N-acetylglutamate (NacGlu), L-tryptophan (L-Trp), 5-HIAA, N-acetyltryptophan (NALT), L-tyrosine (L-Tyr) in cerebrum of *Coq9^+/+* and *Coq9^R239X* mice.
F    Biosynthetic pathway of GSH, serotonin, and catecholamine.

Data information: Data are expressed as mean ± SD. *$P < 0.05$; **$P < 0.01$; ***$P < 0.001$; *Coq9^R239X* mice versus *Coq9^+/+* mice (*t*-test; *n* = 5 for each group).
Source data are available online for this figure.

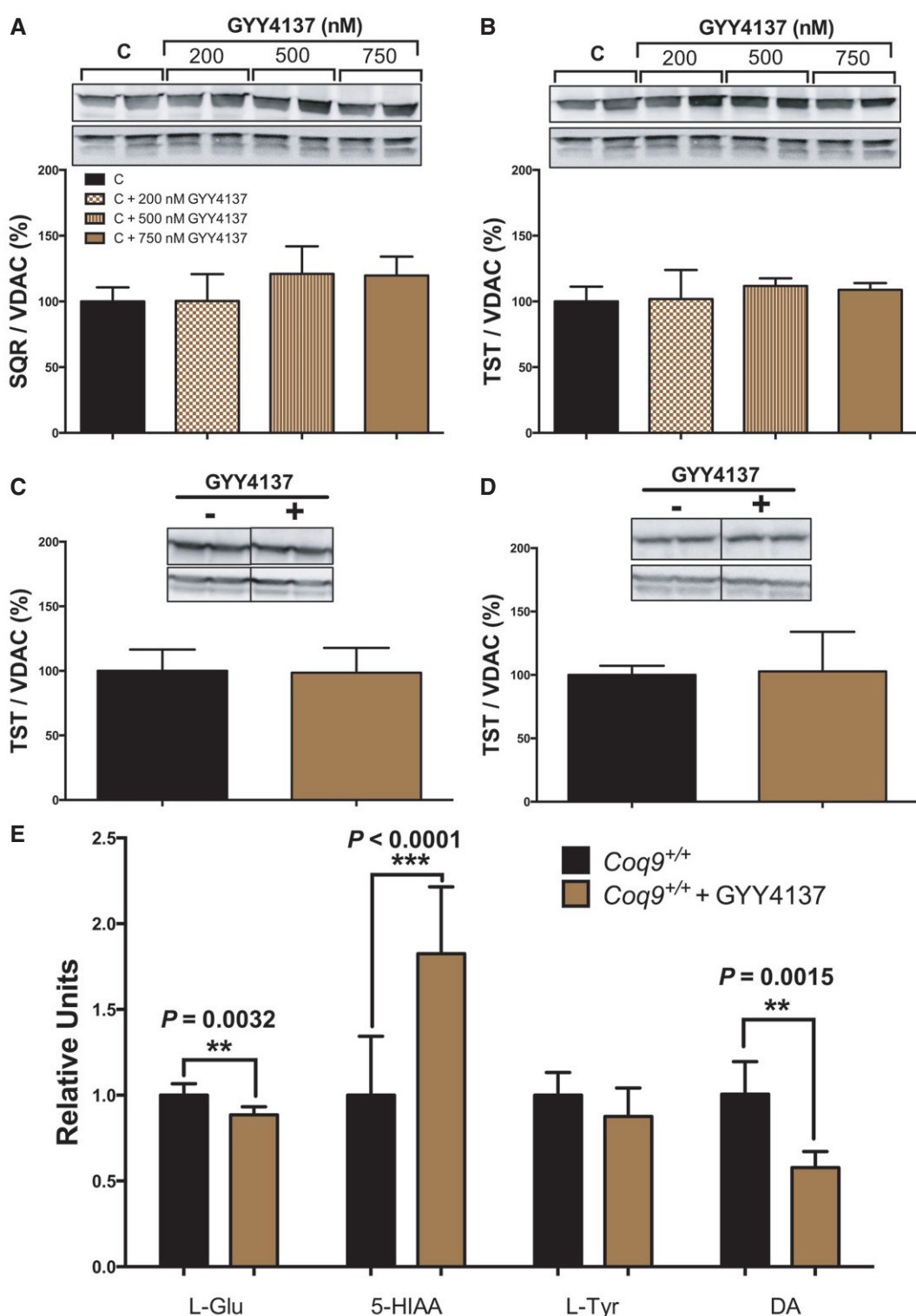

**Figure 7. Supplementation with an H₂S donor in wild-type animals induces changes in neurotransmitters levels.**

A, B   SQR (A) and TST (B) protein levels in human skin fibroblasts supplemented with the H₂S donor GYY4137.
C, D   TST protein level in kidneys (C) and cerebrum (D) of *Coq9*^+/+ mice supplemented with the H₂S donor GYY4137.
E      Levels of neurotransmitters in cerebrum of *Coq9*^+/+ mice supplemented with the H₂S donor GYY4137.

Data information: Images in panels (A and B) were obtained from the same membrane after stripping and re-blotting. Data are expressed as mean ± SD. **$P < 0.01$;
***$P < 0.001$; *Coq9*^+/+ mice supplemented with GYY4137 versus *Coq9*^+/+ mice (t-test; $n = 4$–6 for each group).
Source data are available online for this figure.

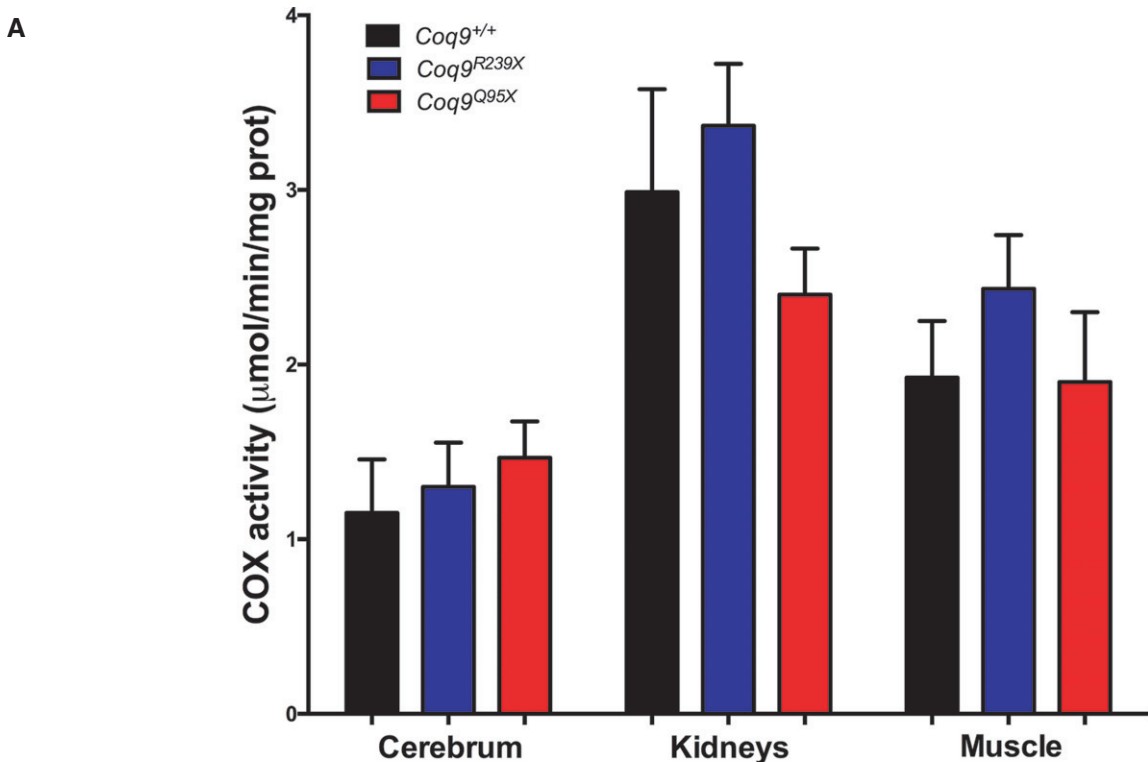

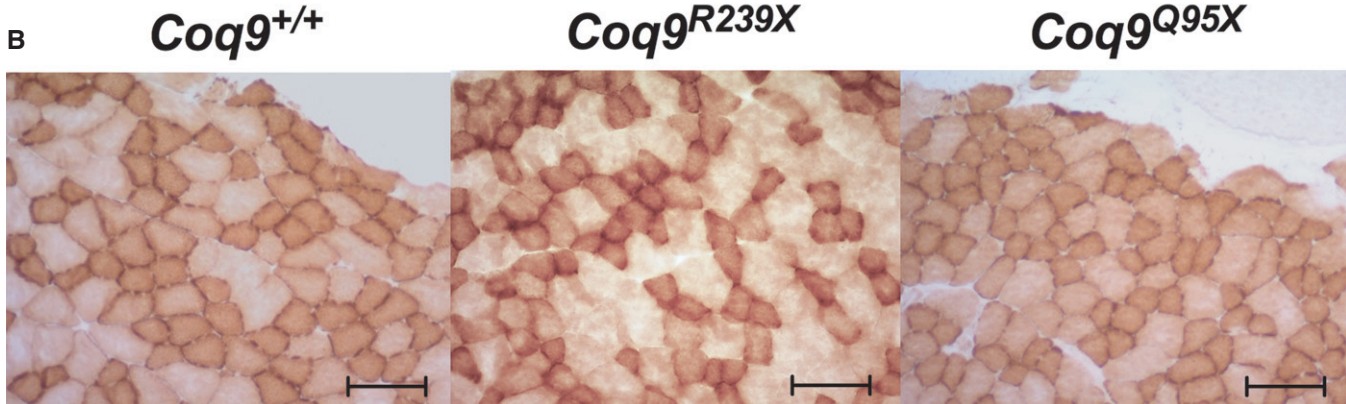

**Figure 8.  COX activity in tissues from CoQ-deficient mice.**

A    COX activity in cerebrum, kidneys, and muscle of *Coq9^{+/+}*, *Coq9^{R239X}*, and *Coq9^{Q95X}* mice. Data are expressed as mean ± SD (one-way ANOVA with a Tukey's *post hoc* test; *n* = 3–6 for each group).

B    Images COX histochemistry in gastrocnemius of *Coq9^{+/+}*, *Coq9^{R239X}*, and *Coq9^{Q95X}* mice; scale bar: 100 μm.

into the mitochondrial electron transport chain (Hildebrandt & Grieshaber, 2008; Modis *et al*, 2013). Low levels of CoQ may thus affect the activity of SQR and its downstream reactions. Here, we demonstrate that severe CoQ deficiency causes a dramatic reduction in SQR levels and activity, which lead to an alteration of the mitochondrial sulfide metabolism. This pattern was observed in a mouse model of primary CoQ deficiency as well as in skin fibroblasts of patients with primary CoQ deficiency due to mutations in different CoQ biosynthetic genes. The deficit in SQR induces changes in the mitochondrial sulfide oxidation pathway with modifications in the levels

of thiols. As a result, biosynthetic pathways of some neurotransmitters were altered in the cerebrum and the blood pressure was reduced. Therefore, this study reveals the reduction in SQR activity as one of the pathomechanisms associated with the CoQ deficiency syndrome.

**Low levels of CoQ induce a disruption in the mitochondrial hydrogen sulfide oxidation pathway**

In this work, we first investigated the tissue levels of *Sqr* mRNA and SQR protein, as well as the SQR activity, in two mouse models of

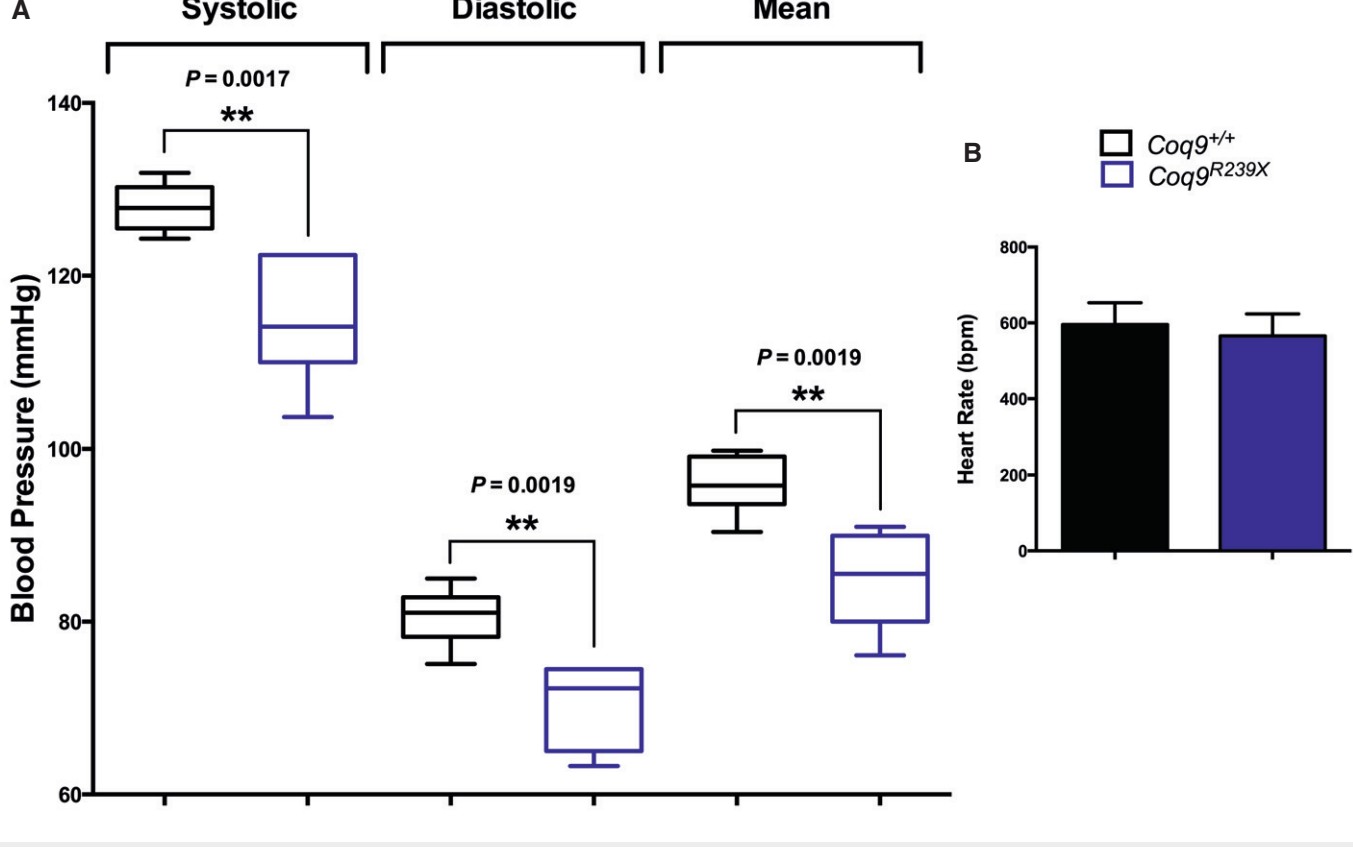

**Figure 9. Blood pressure and heart rate in *Coq9^R239X* mice.**

A   Systolic, diastolic, and mean blood pressure in *Coq9^+/+* and *Coq9^R239X* mice.
B   Heart rate in *Coq9^+/+* and *Coq9^R239X* mice.

Data information: Data are expressed as mean ± SD. **$P < 0.01$; *Coq9^R239X* mice versus *Coq9^+/+* mice (*t*-test; $n = 5$ for each group).

primary CoQ deficiency: the *Coq9^R239X* mouse model with fatal mitochondrial encephalopathy and the *Coq9^Q95X* mouse model with late-onset mild mitochondrial myopathy (Garcia-Corzo *et al*, 2013; Luna-Sanchez *et al*, 2015). Our results confirm that the abundance of the SQR protein is high in kidney, medium–low in muscle, and very low in cerebrum (Geiger *et al*, 2013; http://pax-db.org/protein/2093754/Sqrdl). However, the pattern of response to CoQ deficiency is similar in the three tissues; that is, SQR protein levels and SQR activity in cerebrum, kidneys, and muscle show a correlation between the severity of CoQ deficiency and the decrease in SQR protein levels and activity. Moreover, the deficit on SQR is not limited to mutations in the *Coq9* gene because similar results were obtained in human skin fibroblasts with mutations in different CoQ biosynthetic genes, that is, *PDSS2, COQ2, COQ4,* and *COQ9*. In the mutant fibroblasts, CoQ$_{10}$ depletion causes a significant impairment of SQR-driven respiration, compared with control fibroblasts (Ziosi *et al*, 2017). These results also point out that the deficit in SQR occurs in human primary CoQ$_{10}$ deficiency. Moreover, exogenous supplementation with CoQ$_{10}$ both *in vitro* and *in vivo*, in human and mouse, increased the SQR levels in the mutant cells and mice. The increase in SQR protein levels correlate with the increase in SQR-driven respiration in presence of 5 μM of CoQ$_{10}$ (Ziosi *et al*, 2017). This confirms that the levels of SQR depend on the levels of

CoQ and that exogenous CoQ$_{10}$ supplementation may normalize SQR levels and activity in patients with primary CoQ$_{10}$ deficiency.

As a consequence of the reduced SQR levels, TST activity was increased in cerebrum and kidneys. Interestingly, the cerebral TST activity in *Coq9^R239X* mice was increased at a higher magnitude (2.3-fold increase) than the renal TST (1.3-fold increase), compared in both cases with TST activity in control animals. This response is not likely due to an increase in the hydrogen sulfide because the administration of the H$_2$S donor GYY4137 did not increase TST levels in control fibroblasts and wild-type mice. However, control cells have normal SQR activity that can metabolize hydrogen sulfide. Therefore, we cannot discard that such increases in TST enzyme in *Coq^R239X* mice may be a response to the high levels of hydrogen sulfide, which probably lead to an increase in protein sulfhydration (Mustafa *et al*, 2009; Gao *et al*, 2015). As a consequence, the function of proteins that can be regulated by this posttranslational modification would be affected, and the expression of enzymes potentially involved in the removal of persulfide groups, such as sulfurtransferases, might be induced. Similarly, the levels of SO were only increased in cerebrum of *Coq9^R239X* compared to wild-type animals. Together, these data point out that the disturbances in the mitochondrial hydrogen sulfide oxidation pathway are more accentuated in the cerebrum of the encephalopathic *Coq9^R239X* mouse model.

**Cerebrum of *Coq9^{R239X}* mice shows depletion in the glutathione system: a possible connection with the disruption in the mitochondrial hydrogen sulfide oxidation pathway**

Thiosulfate sulfurtransferase uses GSH and GSH metabolites in its reaction. Therefore, changes in TST activity may induce alterations in GSH metabolism. Moreover, GSH requires cysteine in its biosynthesis, and this amino acid needs hydrogen sulfide for its biosynthesis in fission yeasts (Brzywczy *et al*, 2002). In these organisms, the addition of cysteine reduces the hydrogen sulfide production with CoQ deficiency, suggesting that hydrogen sulfide and cysteine biosynthetic pathways are coordinately regulated by feedback mechanisms (Zhang *et al*, 2008). While the sources of cysteine are apparently different in mammals, the therapeutic effects of NAC, a prodrug of L-cysteine utilized to increase GSH, in *ETHE1* knockout mice suggest that mitochondrial sulfide oxidation pathway may exert some influence on GSH metabolism (Viscomi *et al*, 2010). In fact, sulfur amino acid restriction increased expression of the enzyme cystathionine γ-lyase in mouse liver, resulting in increased hydrogen sulfide production and decreased levels of GSH (Hine *et al*, 2015). Our results showing depletion in the glutathione system in cerebrum would validate this premise, confirming one of the cytotoxic effects of hydrogen sulfide (Truong *et al*, 2006). However, when we checked the direct relation between SQR and GSH systems by transient silencing of SQR mRNA in Hepa cells, we did not obtain a positive correlation. To test a long-term, but moderate, deficiency in SQR, we measured GSH levels in human skin fibroblasts with primary CoQ deficiency. However, also GSH levels were normal in those cases. Thus, the GSH depletion in the cerebrum, kidney, and muscle might be unrelated to the SQR deficiency or, if there is any relation, this one should be linked with a long-term severe SQR deficiency with increased TST activity because the latter enzyme might play a key role in GSH metabolism (Remelli *et al*, 2012). Alternatively, the depletion in the glutathione levels may be related to a reduction in its precursors, for example, cerebral levels of L-glutamate, an essential aminoacid for GSH biosynthesis, were decreased in *Coq9^{R239X}* mice. Moreover, it is important to note that the depletion in GSH levels ran in parallel to a reduction in the levels and activity of the GSH-utilizing enzymes GPx4 and GRd and these changes may be critical for the increase in oxidative damage, neural death, and astrogliosis observed in the cerebrum of *Coq9^{R239X}* mice (Seiler *et al*, 2008; Yoo *et al*, 2012; Garcia-Corzo *et al*, 2013).

**Pathophysiological consequences of SQR deficiency**

The changes in neurotransmitters biosynthesis in the cerebrum of *Coq9^{R239X}* mice are not limited to the reduction in L-glutamate but also to the biosynthesis of serotonin and catecholamines. These alterations might be a consequence of the disruption in hydrogen sulfide metabolism because a dose-dependent and time-dependent decrease in glutamate levels and increase in serotonin levels have been described in the cerebrum and frontal cortex of rats chronically exposed to 20 ppm and 75 ppm of hydrogen sulfide (Skrajny *et al*, 1992; Roth *et al*, 1995). The same authors reported a decrease in the levels of norepinephrine with the exposure to 20 ppm of hydrogen sulfide and an increase in those levels with the exposure to 75 ppm of hydrogen sulfide (Skrajny *et al*, 1992). Norepinephrine and epinephrine levels were, however, decreased in isolated porcine iris-

ciliary body exposed to increased concentrations of NaSH, which is a commonly used hydrogen sulfide donor (Kulkarni *et al*, 2009). The changes in the levels of L-glutamate, 5-HIAA, and dopamine observed in wild-type animals after 2 weeks of supplementation with the H$_2$S donor GYY4173 confirm the influence of hydrogen sulfide in the levels of some neurotransmitters. Thus, the changes in the biosynthesis of amino acid neurotransmitters may contribute to the encephalopathy reported in patients and mice with fatal *Coq9* mutations (Duncan *et al*, 2009; Garcia-Corzo *et al*, 2013; Luna-Sanchez *et al*, 2015; Danhauser *et al*, 2016) and add new evidences about the potential role of hydrogen sulfide as endogenous neuromodulator (Eto *et al*, 2002a,b). Curiously, multiple system atrophy, a neurodegenerative disorder related to disruption in catecholamines metabolism, neuronal death, and astrogliosis, and clinically manifested with problems in movement and autonomic functions of the body such as blood pressure regulation, has been recently associated with mutations in *COQ2* (Multiple-System Atrophy Research Collaboration, 2013), a gene involved in CoQ biosynthesis (Ashby *et al*, 1992; Uchida *et al*, 2000; Quinzii *et al*, 2006).

In the cardiovascular system, hydrogen sulfide regulates smooth muscle contractility (Kabil *et al*, 2014). Generally, hydrogen sulfide induces smooth muscle relaxation, but under particular conditions, it may induces vasoconstriction (Hosoki *et al*, 1997; Kabil *et al*, 2014). Mice in which the *Cse* gene is disrupted exhibit hypertension in comparison with wild-type animals (Yang *et al*, 2008), although in a second and independent study, *Cse* KO mice were reported to be normotensive (Ishii *et al*, 2010). Our results showing a decrease in blood pressure in *Coq9^{R239X}* mice would validate the role of hydrogen sulfide in smooth muscle relaxation. However, we cannot exclude that the decrease in blood pressure could also be the result of the severe dysfunction in the brainstem (Garcia-Corzo *et al*, 2013), where the cardiac and vasomotor centers are localized to regulate the autonomic functions of heart rate and blood pressure. Nevertheless, in contrast to the *Ndufs4* KO mice, another mouse model of Leigh syndrome in which a low heart rate was associated with the brainstem pathology (Quintana *et al*, 2012), the heart rate was normal in *Coq9^{R239X}* mice.

Finally, it has been also reported that hydrogen sulfide and/or its metabolites can interfere with COX activity (Kabil *et al*, 2014; Szabo *et al*, 2014). Our results did not show a decrease in COX activity in the mutant mice, as it has been reported in ethylmalonic encephalopathy due to mutations in *ETHE1* (Tiranti *et al*, 2009). However, the increase in hydrogen sulfide in *ETHE1* mice was higher than in *Coq9^{R239X}* mice, compared in both cases with hydrogen sulfide levels in wild-type animals. Moreover, *ETHE1* mice accumulate thiosulfate, while *Coq9^{R239X}* mice do not (Tiranti *et al*, 2009; Di Meo *et al*, 2011). Therefore, hydrogen sulfide and/or thiosulfate must reach critical levels in order to produce toxic effects on COX activity.

**Conclusions**

Our study demonstrates that primary CoQ deficiency is associated with a disruption of the mitochondrial hydrogen sulfide oxidation pathway, which may be a new pathomechanism associated with this syndrome and may contribute to explain its clinical heterogeneity. According to that, primary CoQ deficiency may be considered the first disease associated with a defect in SQR and, together with

ETHE1 and SO deficiencies (Mudd *et al*, 1967; Garrett *et al*, 1998; Mineri *et al*, 2008; Tiranti *et al*, 2009), the third disease occurring with a defect in the mitochondrial hydrogen sulfide oxidation pathway. Moreover, this new pathomechanism should be taken into consideration for the treatment of primary CoQ deficiency and the evaluation of new experimental therapies.

# Materials and Methods

### Mouse models and treatments

The $Coq9^{R239X}$ and $Coq9^{Q95X}$ mouse models were previously generated and characterized under mix of C57BL/6N and C57BL/6J genetic backgrounds (Garcia-Corzo *et al*, 2013; Luna-Sanchez *et al*, 2015). $Coq9^{R239X/+}$ mice were crossbreed in order to generate $Coq9^{+/+}$, $Coq9^{R239X/+}$, and $Coq9^{R239X/R239X}$ (referred in the article to as $Coq9^{R239X}$). $Coq9^{Q95X/+}$ mice were crossbreed in order to generate $Coq9^{+/+}$, $Coq9^{Q95X/+}$, and $Coq9^{Q95X/Q95X}$ (referred in the article to as $Coq9^{Q95X}$). Only homozygous wild-type and mutant mice between 3 and 5 months of age were used in the study. $Coq9^{R239X}$ mice were treated with ubiquinol-10 ($Q_{10}H_2$) in the drinking water in a dose of 240 mg/kg bw/day during 2 months, as previously reported (Garcia-Corzo *et al*, 2014). $Coq9^{+/+}$ mice were treated with the $H_2S$ donor GYY4137 in the drinking water in a dose of 50 mg/kg bw/day during 2 weeks (Hine *et al*, 2015). Animals were genotyped and randomly assigned in experimental groups. Equal number of males and females were assigned in each experimental group, and no sex differences were observed in the results. A total number of 152 mice were used in this study, excluding breeders and heterozygous mice obtained in each litter.

Mice were housed in the Animal Facility of the University of Granada under an SPF zone with lights on at 7:00 AM and off at 7:00 PM. Mice had unlimited access to water and rodent chow. All experiments were performed according to a protocol approved by the Institutional Animal Care and Use Committee of the University of Granada (procedures 92-CEEA-OH-2015) and were in accordance with the European Convention for the Protection of Vertebrate Animals used for Experimental and Other Scientific Purposes (CETS # 123), the directive 2010/63/EU on the protection of animals used for scientific purposes and the Spanish law (R.D. 53/2013).

### Cell culture and treatments

Primary mutant and control fibroblasts were grown in high glucose DMEM-GlutaMAX medium supplemented with 10% FBS, 1% MEM non-essential amino acids, and 1% antibiotics/antimycotic at 37°C and 5% $CO_2$. Control and mutant fibroblasts were treated with 5 μM $CoQ_{10}$ during 1 or 7 days, as previously reported (Lopez *et al*, 2010). Control fibroblasts were treated with different concentrations of the $H_2S$ donor GYY4137 for 5 days, as published elsewhere (Lee *et al*, 2011).

The Hepa1c1c7 murine hepatoma cell line was obtained from cell bank of the University of Granada and maintained in MEMα GlutaMAX with 10% FBS and 1% antibiotics/antimycotics at 37°C and 5% $CO_2$.

### Subcellular fractionation

Mitochondrial isolation was performed as previously described (Fernandez-Vizarra *et al*, 2002). Tissues were homogenized in a glass–Teflon homogenizer. Kidney was homogenized (1:4, w/v) in the homogenization medium A (0.32 M sucrose, 1 mM EDTA, 10 mM Tris–HCl, ph 7.4); cerebrum was homogenized (1:5, w/v) in the homogenization medium A plus 0.2% free fatty acids BSA, and skeletal muscle was homogenized (1:20, w/v) with Ultraturex in homogenization medium C (0.12 M KCl, 0.02 M HEPES, 2 mM $MgCl_2$, 1 mM EGTA, 5 mg/ml free fatty acids BSA). Kidney and cerebrum homogenates were centrifuged at $1,000 \times g$ for 5 min at 4°C to remove nuclei and debris. Cytosol was collected from supernatants after centrifuging at $14,400 \times g$ for 2 min at 4°C and stored at −80°C after the addition of Halt™ protease and phosphatase inhibitor cocktail (ThermoFisher). The mitochondrial pellet was washed and suspended in homogenization medium and centrifuged again for $14,400 \times g$ for 2 min at 4°C. The final crude mitochondrial pellet was store at −80°C. Skeletal muscle homogenate was centrifuged at $600 \times g$ for 10 min at 4°C. The supernatant (s1) was kept on ice, and the pellet was re-suspended in 8 volumes of buffer A and centrifuged at $600 \times g$ for 10 min at 4°C. The subsequent supernatant (s2) was combined with s1 and centrifuged at $17,000 \times g$ for 10 min at 4°C. The cytosolic supernatant obtained was stored at −80°C after the addition of Halt™ protease and phosphatase inhibitor cocktail (ThermoFisher), and the pellet obtained was re-suspended in 10 volumes of medium A and centrifuged at $7,000 \times g$ for 10 min at 4°C. The pellet was re-suspended in 1 volume of medium B (0.3 M sucrose, 2 mM HEPES, 0.1 mM EGTA) and centrifuged at $3,000 \times g$ for 10 min at 4°C.

### Quantification of $CoQ_{10}$ levels in human skin fibroblasts

After lipid extraction from homogenized cultured skin fibroblasts, $CoQ_{10}$ was determined via reversed-phase HPLC coupled to electrochemical (EC) detection (Lopez *et al*, 2010). The results were expressed in ng $CoQ_{10}$/mg protein.

### Gene expression analyses

Total cellular RNA from frozen tissue was extracted and electrophoresed in a 1.5% agarose gel to check the RNA integrity. RNA from muscle and cerebrum samples was extracted with RNeasy Fibrous Tissue Midi kit (for muscle) and RNeasy Lipid Tissue Mini kit (for cerebrum) (Qiagen, Hilden, Germany) and treated with RNase-Free DNase (Qiagen). RNA from kidney samples was extracted with Real Total RNA Spin Plus Kit (Real). Total RNA was quantified by optical density at 260/280 nm and was used to generate cDNA with High Capacity cDNA Reverse Transcription Kit (Applied Biosystems). Amplification was performed with quantitative real-time PCR, by standard curve method, with specific Taqman probes (from Applied Biosystems) for the targeted gene mouse *Sqrdl* (Mm00502443_m1) and the mouse *Hprt* probe as a standard loading control (Mm01545399_m1) (Luna-Sanchez *et al*, 2015).

### Sample preparation and Western blot analysis in cells

For Western blot analyses in cerebrum, kidney, and muscle, samples were homogenized in T-PER® buffer (Thermo Scientific)

with protease inhibitor cocktail (Pierce) at 1,100 rpm in a glass–Teflon homogenizer. Homogenates were sonicated and centrifuged at 1,000 × *g* for 5 min at 4°C, and the resultant supernatants were used for Western blot analysis. For Western blot analyses in cerebral mitochondria, the pellets containing the mitochondrial fraction were re-suspended in RIPA buffer with protease inhibitor cocktail. For Western blot analyses in skin fibroblasts, cells were collected, washed twice with 1× PBS, and homogenized in RIPA buffer with protease inhibitor cocktail. Homogenates were centrifuged at 14,000 × *g* for 15 min at 4°C, and the resultant supernatants were used for Western blot analysis. 60 μg of protein from the sample extracts was electrophoresed in 12% Mini-PROTEAN TGX™ precast gels (Bio-Rad) using the electrophoresis system mini-PROTEAN Tetra Cell (Bio-Rad). Proteins were transferred onto PVDF 0.45-μm membranes using a mini Trans-blot Cell (Bio-Rad) or Trans-blot Cell (Bio-Rad) and probed with target antibodies. Protein–antibody interactions were detected with peroxidase-conjugated horse anti-mouse, anti-rabbit, or anti-goat IgG antibodies using Amersham ECL™ Prime Western Blotting Detection Reagent (GE Healthcare, Buckinghamshire, UK). Band quantification was carried out using an Image Station 2000R (Kodak, Spain) and a Kodak 1D 3.6 software. Protein band intensity was normalized to VDAC1 (mitochondrial proteins) or GAPDH, and the data expressed in terms of percent relative to wild-type mice or control cells (Luna-Sanchez *et al*, 2015).

The following primary antibodies were used: anti-SQRDL (Proteintech, 17256-1-AP), anti-TST (Proteintech, 16311-1-AP), Anti-SUOX (Proteintech, 15075-1-AP), anti-ETHE1 (Sigma, HPA029029), anti-GPx4 (Abcam, ab125066), anti-GRd (Santa Cruz Biotechnology, sc-32886), anti-VDAC1 (Abcam, ab14734), and anti-GAPDH (Santa Cruz Biotechnology, sc-166574).

### Histochemical analysis of COX activity

Muscle samples (gastrocnemius) were freed from excess connective tissue, embedded in OCT compound (Tissue-Tek), and oriented so that fibers could be cut transversely. Samples then were snap-frozen in precooled isopentane in liquid nitrogen. 8-μm-thick cryosections were placed on Superfrost microscope slides at −20°C by using a Leica CM1510S Cryostat and stained for detection of COX activity as described previously (Tanji & Bonilla, 2008; Luna-Sanchez *et al*, 2015). The sections were examined, and digital images were acquired using a Carl Zeiss Primo Star Optic microscope and a Magnifier AxioCam ICc3 digital camera.

### Enzymatic activities

Sulfide:quinone oxidoreductase activity was determined in isolated mitochondria by following the enzymatic reduction rate of decylubiquinone at 275 nm upon sulfide addition (100 μM) (Hildebrandt & Grieshaber, 2008; Theissen & Martin, 2008).

Thiosulfate sulfurtransferase activity was determined in isolated mitochondria by measuring the formation of thiocyanate from cyanide and thiosulfate (Sorbo, 1955; Hildebrandt & Grieshaber, 2008).

Cytochrome oxidase activity was measured in isolated mitochondria following the reduction in cytochrome C (cyt C) at 550 nm (DiMauro *et al*, 1987).

Glutathione peroxidase (GPx) and reductase (GRd) activities were measured spectrophotometrically from cytosolic fractions following the NADPH oxidation for 3 min at 340 nm on a Shimadzu UV spectrophotometer (UV-1700; Duisburg, Germany) (Griffith, 1999; Lopez *et al*, 2006). The enzyme activities were expressed as nmol/min/mg protein. In both cases, non-enzymatic NADPH oxidation was subtracted from the overall rates.

### Measurement of sulfides, thiosulfate, and sulfite

The concentrations of sulfite and thiosulfate were measured using the monobromobimane HPLC method (Hildebrandt & Grieshaber, 2008). Additionally, hydrogen sulfide release was measured in approximately 100 mg fresh tissue homogenate in passive lysis buffer (Promega) supplemented with 10 mM Cys and 8 mM PLP. A lead acetate hydrogen sulfide detection paper (Sigma) was placed above the liquid phase in a closed Eppendorf tube and incubated for 5 h at 37°C until lead sulfide darkening of the paper occurred (Hine *et al*, 2015).

### GSH measurement

Glutathione measurements were performed in cytosol and mitochondrial fractions of mouse tissues, as well as in Hepa1c1c7 cells and human skin fibroblasts, which were cultured in Opti-MEM medium lacking FBS for 72 h in order to avoid influence of the GSH contained in the FBS.

Glutathione was measured by an established fluorometric method (Hissin & Hilf, 1976). Mitochondrial pellets were resuspended in sodium phosphate buffer (A) (100 mM sodium phosphate, 5 mM EDTA-Na$_2$, pH 8.0). Mitochondrial and cytosolic fractions were deproteinized with ice-cold TCA 40% and centrifuged at 20,000 *g* for 15 min. For GSH measurement, the supernatant was incubated with a solution of (A) and orthophthalaldehyde/ethanol (B) (1 mg/ml) for 15 min at room temperature. The fluorescence of the samples was then measured at 340 nm excitation and 420 nm emission wavelengths in a spectrofluorometer plate reader (Bio-Tek Instruments Inc., Winooski, VT, USA). For GSSG measurement, the supernatant was preincubated with N-ethylmaleimide solution (5 mg/ml) for 40 min and then alkalinized with 0.1 N NaOH (C). Aliquots of these mixtures were then incubated with (B) and (C) for 15 min at room temperature. The fluorescence was then measured as before. GSH and GSSG concentrations were calculated according to standard curves prepared, and the levels of GSH and GSSG are expressed in nmol/mg protein.

### Metabolite quantification in the cerebrum

To quantify the metabolites in the cerebrum of *Coq9$^{+/+}$* and *Coq9$^{R239X}$* mice, frozen samples (−80°C) were lyophilized (Virtis-Benchtop K, Fisher Scientific, Spain) previous to sample homogenization and metabolite extraction. The extraction/homogenization method was adapted from Romisch-Margl *et al* (2012). Briefly, metabolites were extracted from liver and brain samples by adding 300 μl of a mixture of precooled methanol/water (8:2, v/v) to ~50 mg of lyophilized sample and extracted/homogenized in 1.5-ml microcentrifugation tubes with 1.4-mm stainless steel beads using a bead-beating homogenizer (Bullet blender blue, Next Advance,

USA) equipped with an integrated cooler unit. The tissues were homogenized three times for 30 s at 6,000 rpm. Afterward, the tubes were bath-sonicated for 1 min, incubated at 4 °C for 60 min, and centrifuged at 5,000 × $g$ for 15 min at 4°C. Supernatants were analyzed by LC-(ESI)qTOF.

LC-HRMS analyses were performed according to Agilent METLIN/PCDL method [2] using a 1290 infinity UHPLC coupled to a 6550 ESI-QTOF (Agilent Technologies, USA) operated in positive and negative electrospray ionization mode. Briefly, metabolites were separated on Zorbax SB-Aq RR (50 × 2.1 mm., 1.8 μm) column using a continuous gradient elution.

LC-HRMS data were deconvoluted using Find by Molecular Feature algorithm from Mass Hunter Qualitative analysis software (Agilent Technologies), and detected features were aligned across samples using the Mass Profiler Professional (MPP) software (Agilent Technologies). Relative quantification of metabolites was based on peak area of each feature normalized by sample weight. Metabolite identification was performed by using METLIN/PCDL database with the ID browser extension from MPP software, which combines retention time with accurate mass matching to provide greater confidence in compound identification.

To figure out whether an increase in H$_2$S was responsible for the changes in neurotransmitters levels, L-glutamate, 5-HIAA, L-tyrosine, and dopamine were quantified in cerebrum of $Coq9^{+/+}$ mice and $Coq9^{+/+}$ mice supplemented with GYY4137. The quantification was performed by UHPL–CMS/MS following the method described by Santos-Falinda and colleagues (Santos-Fandila et al, 2015).

### Silencing SQR

SiRNA oligonucleotides, Silencer Select Pre-designed siRNAs (Life Technologies, Carlsbad, CA, USA), were used for the transient silencing of SQR. For SQR, the applied sequence contained the 5′-3′ sense, GCUCAGUAAACAUCCCGUUtt and antisense 3′-5′, AACGGGAUGUUUACUGAGCca. This SQR-silencing siRNA (Ambion s81773) was complementary with the mRNAs belonging to RefSeq NM_001162503.1 and NM_021507.5 genes and also targeted exon 3. For negative control, non-targeting siRNAs were applied with the same chemical modifications for enhanced efficacy as in other Silencer Select siRNAs (Ambion). Silencing was conducted as described previously (Modis K, Faseb 2012). Cells (80,000 cells/well) were seeded into 6-well tissue culture plates and cultured in normal culture medium to reach 50% confluence. At this point, the growth medium was replaced with Opti-MEM medium lacking FBS and antibiotics/antimicotics, followed by transfection with 25 pmol siRNA fragments per well at 30 nM forming complexes with 7.5 ml of Lipofectamine® RNAiMAX (Life Technologies). Control cells were transfected in parallel with non-targeting siRNA (Life Technologies).

### Blood pressure and heart rate measurements in mice

Systolic blood pressure (SBP) and heart rate (HR) were measured in conscious, prewarmed, and restrained mice by tail-cuff plethysmography (Digital Pressure Meter LE 5001, Letica S.A., Barcelona, Spain) as described previously (Gomez-Guzman et al, 2014). Briefly, mice were held in a plastic tube, and their tail was put through a rubber cuff, and the cuff was inflated with air. The

### The paper explained

**Problem**
Coenzyme Q (CoQ) has several functions in the cellular metabolism. These functions may be differentially altered under CoQ deficiency, resulting in different clinical presentations. So far, reductions in mitochondrial bioenergetics, pyrimidine biosynthesis, and β-oxidation, as well as increases in oxidative damage and apoptosis, have been described as pathomechanisms of CoQ deficiency syndrome.

**Results**
This study demonstrates that the severity of CoQ deficiency correlates with the decrease in sulfide:quinone oxidoreductase (SQR) levels and activity. The reduction in SQR activity leads to an alteration on mitochondrial hydrogen sulfide oxidation pathway, which results in a modification in the levels of thiols and a decline in the glutathione system. These changes may contribute to the bioenergetics impairment, the increase in oxidative damage and the neuropathology.

**Impact**
Our study demonstrates that primary CoQ deficiency is associated with a disruption of the mitochondrial hydrogen sulfide oxidation pathway, a new pathomechanism associated with this syndrome. These findings should be taken into consideration for the treatment of primary CoQ deficiency and the evaluation of new experimental therapies. The results shown in this article also consolidate the defects in sulfide oxidation as a group of mitochondrial diseases. Furthermore, the data shown here have important implications to elucidate the role of hydrogen sulfide in the regulation of COX activity, the modulation of blood pressure, and its actions as neuromodulator.

pressure level at which the first pulse appeared, after blood flow had been interrupted with the inflated cuff, was designated SBP. At least fifteen determinations were made in every session, and the mean of the lowest ten values within 5 mmHg was taken as the SBP level. HR values were obtained as average of several determinations simultaneously to SBP level.

### Statistical analysis

All statistical analyses were performed using the Prism 6 scientific software. Data are expressed as the mean ± SD of four–six experiments per group. A one-way ANOVA with a Tukey's post hoc test was used to compare the differences between three experimental groups. Studies with two experimental groups were evaluated using unpaired Student's t-test. A $P$-value of < 0.05 was considered to be statistically significant.

**Expanded View** for this article is available online.

### Acknowledgements

We are grateful to Dr. Iryna Rusanova (Universidad de Granada) for her technical support. We thank to Pol Herrero (Metabolomics Facility at the Center for Omic Sciences (COS) of the University Rovira i Virgili, recognized as a Singular Research and Technology Infrastructure by the Spanish Ministry of Economy and Competitiveness) for his contributions to mass spectrometry analysis. This work was supported by grants from Ministerio de Economía y Competitividad, Spain, and the ERDF (SAF2013-47761-R, SAF2014-55523-R, RD12/0042/0011 and SAF2015-65786-R), from the Consejería de Economía, Innovación, Ciencia y Empleo, Junta de Andalucía (P10-CTS-6133), from the NIH (P01HD080642) and from the

foundation "todos somos raros, todos somos únicos". MLS is a predoctoral fellow from the Consejería de Economía, Innovación, Ciencia y Empleo, Junta de Andalucía. LCL is supported by the "Ramón y Cajal" National Programme, Ministerio de Economía y Competitividad, Spain (RYC-2011-07643).

## Author contributions

ML-S leaded the study, conducted the experiments related to the hydrogen sulfide metabolism, COX activity/staining, and the GSH system, analyzed the results, and wrote the manuscript. AH-G conducted most of the Western blot assays, performed the experimental procedures with the H$_2$S donor, participated in the experiments with Hepa cells and human skin fibroblasts, and analyzed the results. JC-S conducted some Western blot assays and the experiments with Hepa cells. TMH supervised and conducted the enzymatic assays and some HPLC analysis. EB-C conducted some experiments with Hepa cells and human skin fibroblasts. ÁS-F quantified the neurotransmitters after supplementation with GYY4137. RKAS conducted some histochemistry assays. MR and JD performed the analysis of blood pressure and heart rate. HP, MS, and FD provided the human skin fibroblasts from patients with CoQ deficiency. GE and DA-C contributed to the discussion and edited the manuscript. LCL conceived the idea for the project, supervised the experiments, and edited the manuscript. All authors critically reviewed the manuscript.

## Conflict of interest

The authors declare that they have no conflict of interest.

## For more information

Online Mendelian Inheritance in Man (OMIM): PDSS2, COQ2, COQ4, COQ9
http://omim.org/entry/610564?search=pdss2&highlight=pdss2
http://omim.org/entry/609825?search=COQ2&highlight=coq2
http://omim.org/entry/612898?search=COQ4&highlight=coq4
http://omim.org/entry/612837?search=COQ9&highlight=coq9

International mito-patients
http://www.mitopatients.org/

The association of mitochondrial disease patients in Spain
http://www.aepmi.org/publicoIngles/index.php

The United Mitochondrial Disease Foundation
http://www.umdf.org/

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
