## [Review Process File · EMBO Molecular Medicine]

CoQ Deficiency Causes Disruption of Mitochondrial Sulfide Oxidation, a new Pathomechanism Associated with this Syndrome

Marta Luna-Sánchez, Agustín Hidalgo-Gutiérrez, Tatjana M. Hildebrandt, Julio Chaves-Serrano, Eliana Barriocanal-Casado, Ángela Santos-Fandila, Miguel Romero, Ramy K. A. Sayed, Juan Duarte, Holger Prokisch, Markus Schuelke, Felix Distelmaier, Germaine Escames, Darío Acuña-Castroviejo, and Luis C. López

Corresponding author: Luis Lopez, University of Granada

Review timeline:

Submission date:	25 February 2016
Editorial Decision:	14 April 2016
Revision received:	17 June 2016
Editorial Decision:	16 October 2016
Revision received:	17 October 2016
Accepted:	19 October 2016

Transaction Report:

Editor: Roberto Buccione

1st Editorial Decision

14 April 2016

Thank you for the submission of your manuscript to EMBO Molecular Medicine. We are sorry that it has taken longer than usual to get back to you on your manuscript. In this case we experienced some difficulties in securing three appropriate expert reviewers, also due to the necessity to evaluate two back-to-back submissions, and then obtaining their evaluations in a timely manner. Furthermore, one reviewer (#1) ultimately did not deliver.

As you will see the two Reviewers are globally positive, but do raise many issues. Reviewer 3, especially, raises an important and fundamental one. Although I will not dwell into much detail, I would like to highlight the main points.

In fact, while the issues raised by Reviewer 2 are focused on improving precision, quality of controls and clarifying important biochemical aspects, Reviewer 3 feels that, in addition to other items of concern, without improved mechanistic understanding and more conclusive demonstration of causal links, the manuscript would not be suited for publication. Specifically, s/he would like to understand how SQR activity is suppressed by decreased CoQ and see support for the claim that decreased SQR is responsible for diminished brain neurotransmitter levels. I agree that this is the main focus of your work, which should thus be further developed in a mechanistic sense. I should also mention that when deciding whether to send your manuscript out for review, I had sought counsel from an external advisor who had agreed that the manuscript (s) was very interesting but

noted that the potential shortcoming that the mechanisms were not clearly defined.

In conclusion, while publication of the paper cannot be considered at this stage, given the potential interest of your findings, we have decided to give you the opportunity to address the above concerns. We are thus prepared to consider a substantially revised submission, with the understanding that the Reviewers' concerns must be addressed with additional experimentation as appropriate and that acceptance of the manuscript will entail a second round of review.

Please note that it is EMBO Molecular Medicine policy to allow a single round of revision only and that, therefore, acceptance or rejection of the manuscript will depend on the completeness of your responses included in the next, final version of the manuscript.

As you know, EMBO Molecular Medicine has a "scooping protection" policy, whereby similar findings that are published by others during review or revision are not a criterion for rejection. However, I do ask you to get in touch with us after three months if you have not completed your revision, to update us on the status. Please also contact us as soon as possible if similar work is published elsewhere.

EMBO Molecular Medicine now requires a complete author checklist (<http://embomolmed.embopress.org/authorguide#editorial3>) to be submitted with all revised manuscripts. Provision of the author checklist is mandatory at revision stage; The checklist is designed to enhance and standardize reporting of key information in research papers and to support reanalysis and repetition of experiments by the community. The list covers key information for figure panels and captions and focuses on statistics, the reporting of reagents, animal models and human subject-derived data, as well as guidance to optimise data accessibility.

I look forward to seeing a revised form of your manuscript as soon as possible.

***** Reviewer's comments *****

Referee #2 (Comments on Novelty/Model System):

It is important to have used both patients' fibroblasts and mouse models

Referee #2 (Remarks):

This study concerns the effects of Coenzyme Q deficiency on sulfide oxidation that uses CoQ as acceptor of sulfide CoQ oxidoreductase (SQR), linking it to the respiratory chain. The study has been performed both in an animal model of two strains of mice with mutations in Coq9, an enzyme of the biosynthetic pathway of CoQ, and in human fibroblasts from patients with different mutations in CoQ biosynthesis. In the mouse models the SQR protein levels and activity decreased to different extents in different tissues, depending on the extent of CoQ deficiency. In addition in human fibroblasts the decreased SQR protein levels were restored by incubation of the cells with CoQ10. In several cases of decreased SQR there was an increase of thiosulfate sulfur transferase TST, an enzyme catalyzing a subsequent reaction. The level of GSH was also decreased. The study then examines consequences of the primary alteration due to changes in the levels of sulfide as a signaling molecule affecting some neurotransmitters and hence some physiological parameters such as the blood pressure.

This is an important paper that conveys the clear idea that some of the physiological consequences of CoQ deficiency are caused by the specific defect of sulfur metabolism. The study is somewhat poor in the characterization of the sulfide oxidation pathway (only SQR and TST are investigated among the several enzymes involved), while is more detailed on the secondary consequences, such as thiol levels and metabolism and levels of some important metabolites, as well as blood pressure. The discussion is long and elaborate, although I agree that the different points had to be discussed, and indeed they have been discussed properly. Probably dissecting the Discussion into paragraphs with specific headings would help the reader.

Some specific points.

1) The abstract should be more informative, indicating the systems studied (mutant mice, human fibroblasts)

- 2) Introduction, line 7. Biogenetics: you mean bioenergetics?
- 3) P. 5 and Fig. 1. Since SQR mRNA is only slightly or not decreased what is the reason for the strongly decreased protein levels? Is it due to enhanced protein degradation in absence of CoQ substrate?
- 4) P. 5, Coq-mutant mice. The CoQ9 levels have been studied in the organs of the same mice used in this study or is it just literature data?
- 5) P.5. What is the extent of incorporation of CoQ10 in the mitochondria of fibroblasts incubated with CoQ10? Is it enough to restore respiratory activities?
- 6) Perhaps the subsidiary Fig. 1 (sulfide metabolic pathway) should be better incorporated in the main manuscript.

Referee #3 (Comments on Novelty/Model System):

Overall, the study describes an association between COA and SQR levels, but suffers from a complete lack of mechanistic insights. To recommend for publication, I would expect the authors to establish the molecular mechanism by which SQR activity is suppressed by diminished levels of CoQ.

Referee #3 (Remarks):

The authors show a drop in SQR levels and enzyme activity in association with severe CoQ deficiency, along with a concomitant increase in TST activity, changes in levels of specific metabolite levels and drop in blood pressure. Based on this finding, loss of SQR activity is implicated as a basis for some of the pathological manifestations observed in patients with CoQ insufficiency.

General comments:

The authors state in the Abstract (and infer elsewhere) that "CoA deficiency causes a reduction in SQR levels ... which leads to an alteration in mitochondrial sulfide metabolism", and "As a result, biosynthetic pathways of glutamate, serotonin and catecholamines were altered in the brain, and blood pressure was reduced". Despite this claim of causality, evidence is lacking. An alternative interpretation is that CoA deficiency causes a decrease in mitochondrial sulfide metabolism and this triggers a decrease in SQR levels. In either case, no hard evidence is provided in support of a molecular mechanism that links CoA deficiency to decreased SQR levels. Further, while it is possible that decreased SQR is responsible for diminished brain levels of neurotransmitters, Tyr and other species, this claim is highly speculative, and not supported by data in the present manuscript. Experimental evidence is needed to establish these inferred direct causal connections.

Specific comments:

Page 6, line 10: refers to a Table S1, which appears to be missing from the manuscript. Please revise accordingly.

The use of * vs # notation to show statistical significance is confusing. It is not always clear which two data points are being compared. The authors should clarify these by using connecting lines to denote statistical comparisons.

Fig 1: The observed CoA-depletion triggered decrease in SQR protein level is more profound than the decrease in RNA level, and enzymatic activity is even more pronounced (apparently complete loss in muscle). The molecular basis for SQR inactivation should be defined experimentally. Does this large decrease in SQR enzymatic activity (as opposed to protein levels) also occur in the human cell model of SQR deficiency?

Fig 2B: This is a crucial experiment to show direct connection between the CoQ level and SQR expression. However, an increase in SQR levels is only shown after 7 days of treatment. How quickly is SQR protein abundance restored by CoQ supplementation? Is SQR activity concomitantly restored?

Fig 3A, B: The Western blots have very high background signal, high variability and VDAC bands are not clearly separated. This raises concern about the accuracy of quantification. TST protein levels in muscle seem to have been omitted. Is this because the levels are too low for detection?

Fig 3C: Most significant increase in TST activity is observed in cerebrum, but the authors were unable to detect SQR expression or activity in this tissue. Meanwhile the effect is minimal in the kidney and absent from the muscle. Fig 1 shows significant decreases in SQR activity in these tissues. Does this indicate a TST response independent of SQR?

Page 6, para 2: The title "Low SQR activity induces an increase in TST activity" can mislead the

reader to thinking that this relationship is cell-autonomous. Please revise for clarity.

In the Discussion (page 10), the authors suggest that an increase in circulating H₂S may cause the observed increase in TST. This hypothesis can be tested experimentally by increasing blood H₂S levels by treatment with an H₂S donor molecule.

Fig 5C: Surprisingly, depletion of CoQ and SQR activity was associated with only a very modest decrease in GSH levels. For a more complete analysis, the authors should also quantify oxidized GSH (i.e., GSSG) and determine the GSH/GSSG ratio as a better indicator of redox status.

Fig S1: Describes the sulfide oxidation pathway, but omits the key enzyme ETHE1. Please revise the Fig to include ETHE1.

1st Revision - authors' response

17 June 2016

Referee #2

We thank the general positive comments of this reviewer, as well as the useful recommendations to improve the manuscript.

1. *This is an important paper that conveys the clear idea that some of the physiological consequences of CoQ deficiency are caused by the specific defect of sulfur metabolism. The study is somewhat poor in the characterization of the sulfide oxidation pathway (only SQR and TST are investigated among the several enzymes involved), while is more detailed on the secondary consequences, such as thiol levels and metabolism and levels of some important metabolites, as well as blood pressure.*

We agree with the reviewers' comment. We have performed additional experiments to quantify the levels of the other enzymes of the mitochondrial hydrogen sulfide oxidation pathway, SDO (ETHE1) and SO (SUOX), in cerebrum, kidneys and muscle of *Coq9^{+/+}*, *Coq9^{R239X}* and *Coq9^{Q95X}* mice. These results, which are shown in the Figure 4, confirm that cerebrum of *Coq9^{R239X}* mice is the tissue with more accentuated disruption in the mitochondrial hydrogen sulfide oxidation pathway. Moreover, these changes may depend on the low levels of SQR rather than on the accumulation of hydrogen sulfide since supplementation with the H₂S donor GYY4137 did not increase TST *in vitro* or *in vivo* (Fig. 7A-D).

2. *The discussion is long and elaborate, although I agree that the different points had to be discussed, and indeed they have been discussed properly. Probably dissecting the Discussion into paragraphs with specific headings would help the reader.*

We have divided the discussion on four sections with specific headings.

Some specific points.

1. *The abstract should be more informative, indicating the systems studied (mutant mice, human fibroblasts).*

We thank the reviewer for noting the lack of this information in the abstract. We have added the studied systems in the abstract.

2. *Introduction, line 7. Biogenetics: you mean bioenergetics?*

We apologize for the typographical error, which we have corrected.

3. *P. 5 and Fig. 1. Since SQR mRNA is only slightly or not decreased what is the reason for the strongly decreased protein levels? Is it due to enhanced protein degradation in absence of CoQ substrate?*

As the reviewer noted, SQR is only affected at protein level in CoQ deficient models. Previous studies demonstrated that CoQ accept electrons from H₂S in the reaction catalyzed by SQR. Therefore, CoQ is essential for the SQR activity. Our results suggest now that in situations with low levels of CoQ, the SQR protein may be unstable, increasing its degradation. To be more convincing about that, we have supplemented the patient's fibroblasts and *Coq9^{R239X}* mice with exogenous

CoQ₁₀ (Fig. 3). After the treatment, SQR increased both *in vitro* and *in vivo*, confirming that the stability of SQR is strongly dependent of the CoQ levels.

4. *P. 5, Coq-mutant mice. The CoQ9 levels have been studied in the organs of the same mice used in this study or is it just literature data?*

The mentioned CoQ₉ levels were reported by our group in a recent study (Luna-Sanchez et al., 2015), and we have quoted this article in the manuscript. Additionally, we have now added the total CoQ levels in kidneys and muscle of *Coq9*^{+/+}, *Coq9*^{R239X} and *Coq9*^{R239X} + CoQ₁₀ mice (Fig. 3 C-D). These new data may help to clarify the connection between CoQ and SQR.

5. *P.5. What is the extent of incorporation of CoQ10 in the mitochondria of fibroblasts incubated with CoQ10? Is it enough to restore respiratory activities?*

We thank the reviewer for pointing out the inadequate explanation about the incorporation of CoQ₁₀ in the mitochondria of fibroblasts incubated with CoQ₁₀. We have added a new experimental group with CoQ₁₀ supplementation for 1 day (Fig. 2B). Our results show that SQR is increased after 7 days of CoQ₁₀ supplementation but not after 1 day of CoQ₁₀ supplementation. These results correlate with a previous study where we demonstrated that ATP levels were normalized in CoQ₁₀ deficient fibroblast after 7 days of CoQ₁₀ supplementation but not after 1 day (Lopez et al., 2010). We have quoted this article and added a sentence in the results section. Taken together, these data confirm the prolonged pharmacokinetics of CoQ₁₀ to reach the mitochondria.

6. *Perhaps the subsidiary Fig. 1 (sulfide metabolic pathway) should be better incorporated in the main manuscript.*

The previous Figure S1 is now the Figure 1 in this revised version of the manuscript.

Referee #3

We appreciate the insightful comments of this reviewer.

General comments:

1. *The authors state in the Abstract (and infer elsewhere) that "CoA deficiency causes a reduction in SQR levels ... which leads to an alteration in mitochondrial sulfide metabolism", and "As a result, biosynthetic pathways of glutamate, serotonin and catecholamines were altered in the brain, and blood pressure was reduced". Despite this claim of causality, evidence is lacking. An alternative interpretation is that CoA deficiency causes a decrease in mitochondrial sulfide metabolism and this triggers a decrease in SQR levels. In either case, no hard evidence is provided in support of a molecular mechanism that links CoA deficiency to decreased SQR levels. Further, while it is possible that decreased SQR is responsible for diminished brain levels of neurotransmitters, Tyr and other species, this claim is highly speculative, and not supported by data in the present manuscript. Experimental evidence is needed to establish these inferred direct causal connections.*

We thank the reviewer for suggesting us new experiments in order to find clear experimental evidences to support our result's interpretations. We hope that our responses are satisfactory. Note that for our response we assume that "CoA" is "CoQ" in the reviewer's text.

Specific comments:

2. *Page 6, line 10: refers to a Table S1, which appears to be missing from the manuscript. Please revise accordingly.*

Table S1 is included in the supplementary document.

3. *The use of * vs # notation to show statistical significance is confusing. It is not always clear which two data points are being compared. The authors should clarify these by using connecting lines to denote statistical comparisons.*

We have edited our figures and all of them contains now connecting lines. We hope that the figures are clearer now in terms of the statistic analysis.

4. *Fig 1: The observed CoA-depletion triggered decrease in SQR protein level is more profound than the decrease in RNA level, and enzymatic activity is even more pronounced (apparently complete loss in muscle). The molecular basis for SQR inactivation should be defined experimentally. Does this large decrease in SQR enzymatic activity (as opposed to protein levels) also occur in the human cell model of SQR deficiency?*

As the reviewer noted, SQR is only affected at protein level in CoQ deficient models. Previous studies demonstrated that CoQ accept electrons from H₂S in the reaction catalyzed by SQR. Therefore, CoQ is essential for the SQR activity. In situations with low levels of CoQ, the SQR protein may be unstable, increasing its degradation. To be more convincing about that, we have supplemented the patient's fibroblasts and *Coq9^{R239X}* mice with exogenous CoQ₁₀ (Fig. 3). After the treatment, SQR increased both *in vitro* and *in vivo*, confirming that the stability of SQR is strongly dependent of the CoQ levels. Therefore, we believe that our study demonstrate the connection between CoQ and SQR in humans and mice.

5. *Fig 2B: This is a crucial experiment to show direct connection between the CoQ level and SQR expression. However, an increase in SQR levels is only shown after 7 days of treatment. How quickly is SQR protein abundance restored by CoQ supplementation? Is SQR activity concomitantly restored?*

We have added a new experimental group with CoQ₁₀ supplementation for 1 day (Fig. 2B). Our results show that SQR is increased after 7 days of CoQ₁₀ supplementation but not after 1 day of CoQ₁₀ supplementation. These results correlate with a previous study where we demonstrated that ATP levels were normalized in CoQ₁₀ deficient fibroblast after 7 days of CoQ₁₀ supplementation but not after 1 day (Lopez et al, 2010). We have quoted this article and added a sentence in the results section. Taken together, these data confirm the prolonged pharmacokinetics of CoQ₁₀ to reach the mitochondria. We have not included the SQR activity in fibroblasts because our enzymatic method requires a huge amount of isolated mitochondria, so it is not sensitive enough to measure the activity in skin fibroblasts. However, Dr. Quinzii is simultaneously submitting a manuscript for a possible back-to-back publication. In that manuscript, Dr. Quinzii indirectly measures SQR activity in the patients' fibroblasts and, importantly, their results show that SQR activity correlates with the CoQ₁₀ in the patients' fibroblasts. Those results are similar to our data in mice tissues.

6. *Fig 3A, B: The Western blots have very high background signal, high variability and VDAC bands are not clearly separated. This raises concern about the accuracy of quantification. TST protein levels in muscle seem to have been omitted. Is this because the levels are too low for detection?*

We have repeated the western blots of TST to have better images (note that the previous Fig. 3 is the Fig 4 in the revised version of the manuscript). We have now added the TST protein levels in muscle. Moreover, we have performed additional experiments to quantify the levels of the other enzymes of the mitochondrial hydrogen sulfide oxidation pathway, SDO (ETHE1) and SO (SUOX), in cerebrum, kidneys and muscle of *Coq9^{+/+}*, *Coq9^{R239X}* and *Coq9^{Q95X}* mice. These results, which are shown in the Figure 4, confirm that cerebrum of *Coq9^{R239X}* mice is the tissue with more accentuated disruption in the mitochondrial hydrogen sulfide oxidation pathway.

7. *Fig 3C: Most significant increase in TST activity is observed in cerebrum, but the authors were unable to detect SQR expression or activity in this tissue. Meanwhile the effect is minimal in the kidney and absent from the muscle. Fig 1 shows significant decreases in SQR activity in these tissues. Does this indicate a TST response independent of SQR?*

We thank the astute reviewer for noting the discrepancy. As it has been previously reported, the abundance of SQR in mouse cerebrum is very low (Geiger et al, 2013) (<http://pax->

db.org/protein/2093754/Sqrdl). For this reason, we could not detect SQR in cerebral homogenates. Because SQR is localized in mitochondria, we decided to isolate cerebral mitochondria and try to detect the SQR in this enriched fraction. As we show in Figure 2D, we were able to detect SQR in isolated mitochondria, and more important, our results show that SQR levels were severely reduced in cerebrum of *Coq9^{Q95X}* mice. These new results support the idea that TST response is dependent on SQR. In fact, the cerebrum, which has the lowest levels of SQR, shows the most accentuated response in the enzymes of the mitochondrial hydrogen sulfide oxidation pathway (Fig. 4).

8. *Page 6, para 2: The title "Low SQR activity induces an increase in TST activity" can mislead the reader to thinking that this relationship is cell-autonomous. Please revise for clarity.*

With the incorporation of the new results the title is now "Low SQR activity induces changes in the proteins involved in the mitochondrial hydrogen sulfide oxidation pathway".

9. *In the Discussion (page 10), the authors suggest that an increase in circulating H₂S may cause the observed increase in TST. This hypothesis can be tested experimentally by increasing blood H₂S levels by treatment with an H₂S donor molecule.*

We thank the reviewer for suggesting us this interesting experiment. We have supplemented control fibroblasts and mice with the H₂S donor GYY4137. However, TST was not increased in response to the H₂S donor (Fig. 7 B-D), suggesting that the increase in TST was not due to an accumulation of H₂S. Most likely is that the induction TST is a response to the low SQR levels. In fact, we were able to detect SQR in cerebral mitochondria, and more important, our results show that SQR levels were severely reduced in cerebrum of *Coq9^{Q95X}* mice. We describe and discuss these results in the manuscript.

We also used this experimental approach to quantify some neurotransmitters in the cerebrum of control mice after two weeks of supplementation with the H₂S donor GYY4137 (Fig. 7).

Interestingly, our results suggest that the changes observed in the levels of some neurotransmitters in the cerebrum of *Coq9^{R239X}* mice may be due to the increase in H₂S. These results are in agreement of previous studies that are quoted in the discussion.

10. *Fig 5C: Surprisingly, depletion of CoQ and SQR activity was associated with only a very modest decrease in GSH levels. For a more complete analysis, the authors should also quantify oxidized GSH (i.e., GSSG) and determine the GSH/GSSG ratio as a better indicator of redox status.*

We have determined GSSG/GSH ratio and the results are included in Fig S3.

11. *Fig S1: Describes the sulfide oxidation pathway, but omits the key enzyme ETHE1. Please revise the Fig to include ETHE1.*

We apologize because the nomenclature of the enzymes involved in sulfide oxidation pathway is confused. ETHE1 is SDO. Our previous Fig S1 (now Fig 1) includes now the different names used for these enzymes.

REFERENCES

- Geiger T, Velic A, Macek B, Lundberg E, Kampf C, Nagaraj N, Uhlen M, Cox J, Mann M (2013) Initial quantitative proteomic map of 28 mouse tissues using the SILAC mouse. *Mol Cell Proteomics* 12: 1709-1722
- Lopez LC, Quinzii CM, Area E, Naini A, Rahman S, Schuelke M, Salviati L, DiMauro S, Hirano M (2010) Treatment of CoQ(10) deficient fibroblasts with ubiquinone, CoQ analogs, and vitamin C: time- and compound-dependent effects. *PLoS One* 5: e11897
- Luna-Sanchez M, Diaz-Casado E, Barca E, Tejada MA, Montilla-Garcia A, Cobos EJ, Escames G, Acuna-Castroviejo D, Quinzii CM, Lopez LC (2015) The clinical heterogeneity of coenzyme Q10 deficiency results from genotypic differences in the *Coq9* gene. *EMBO Mol Med*

Thank you for the submission of your revised manuscript to EMBO Molecular Medicine and many apologies again for the delay in reaching a decision due in part to reviewer delays, but most crucially, to the glitch in our system that effectively hid your manuscript from me.

We have now received the enclosed reports from the referees that were asked to re-assess it. As you will see the reviewers are now globally supportive and I am pleased to inform you that we will be able to accept your manuscript pending the following final amendments:

- 1) I note there are various duplications in the data set, as confirmed by your provided source data files. I appreciate that these are control lanes repeated in the figures for clarity in correlation with quantifications. However, to avoid future post-publication issues and in the paramount interest of clarity and transparency, I must ask you to clearly declare and explain each of these occurrences in the figure legends.
- 2) I found it difficult to match appropriate control bands to source data in fig. 4C. Please indicate them better in the source data file.
- 3) Please provide the source data as individual files for EACH figure.
- 4) The references to the "SI" items need to be changed to "Appendix Figure S1".... and all current SI files together have to be made one Appendix PDF including a ToC as per the author guidelines.
- 5) If possible, please provide a better quality image for Fig. 2F.
- 6) Please include a running title in the title page.
- 7) The manuscript must include a statement in the Materials and Methods identifying the institutional and/or licensing committee approving the experiments, including any relevant details (like how many animals were used, of which gender, at what age, which strains, if genetically modified, on which background, housing details, etc). We encourage authors to follow the ARRIVE guidelines for reporting studies involving animals. Please see the EQUATOR website for details: <http://www.equator-network.org/reporting-guidelines/improving-bioscience-research-reporting-the-arrive-guidelines-for-reporting-animal-research/>. Please make sure that ALL the above details are reported in the manuscript.

Finally, I would also like to mention that the accompanying manuscript is also undergoing a final minor revision (but will not delay the publication of your manuscript).

Please submit your revised manuscript within two weeks. I look forward to seeing a revised form of your manuscript as soon as possible.

***** Reviewer's comments *****

Referee #2 (Remarks):

The authors have replied in a very detailed way and performed additional experiments to cope with the requests of this reviewer. The manuscript is now very convincing and conveys a clear important message.

Referee #3 (Remarks):

The authors have satisfactorily addressed all prior reviewer's concerns with new findings and revised text. The manuscript now makes a compelling case for CoQ levels as a physiological determinant of the sulfide oxidation pathway activity and enzyme expression.

We appreciate the positive comments from the reviewers and you regarding our manuscript number EMM-2016-06345 entitled “CoQ Deficiency Causes Disruption of Mitochondrial Sulfide Oxidation, a new Pathomechanism Associated to this Syndrome”. We made the required changes. Our specific responses are listed below.

- 1) *I note there are various duplications in the data set, as confirmed by your provided source data files. I appreciate that these are control lanes repeated in the figures for clarity in correlation with quantifications. However, to avoid future post-publication issues and in the paramount interest of clarity and transparency, I must ask you to clearly declare and explain each of these occurrences in the figure legends.*

In the legend of figure 4 we now specify that “Images of Figures 4C, F and I were obtained from the same membrane after stripping and re-blotting”. In the legend of figure 7 we now specify that “Images of Figures 7A and B were obtained from the same membrane after stripping and re-blotting”.

- 2) *I found it difficult to match appropriate control bands to source data in fig. 4C. Please indicate them better in the source data file.*

The loading controls in figures 4G and 4H were different to the ones provided in the source data. These errors were due to the last changes performed in this figure after the resubmission. The correct loading controls were the ones provided in the source data file. Therefore, we have made the loading control modifications in the main figures 4G and 4H. These modifications do not change any result. Additionally, we have now included a note in the source data of each figure indicating the lines that were represented in the main figures.

- 3) *Please provide the source data as individual files for EACH figure.*

We have now included individual source data files for each figure.

- 4) *The references to the "SI" items need to be changed to "Appendix Figure S1".... and all current SI files together have to be made one Appendix PDF including a ToC as per the author guidelines.*

All “SI” figures and tables are now referred as “Appendix Figure S” or “Appendix Table S1” in the text and the table and figures are included in one single pdf.

- 5) *If possible, please provide a better quality image for Fig. 2F.*

SQR has low expression in muscle. All SQR images in muscle are similar.

- 6) *Please include a running title in the title page.*

The running title is now included in the title page.

- 7) *The manuscript must include a statement in the Materials and Methods identifying the institutional and/or licensing committee approving the experiments, including any relevant details (like how many animals were used, of which gender, at what age, which strains, if genetically modified, on which background, housing details, etc). We encourage authors to follow the ARRIVE guidelines for reporting studies involving animals. Please see the EQUATOR website for details: <http://www.equator-network.org/reporting-guidelines/improving-bioscience-research-reporting-the-arrive-guidelines-for-reporting-animal-research/>. Please make sure that ALL the above details are reported in the manuscript.*

We have completed some missing information in the material and methods section according to the Equator guidelines.

Corresponding Author Name: Luis C. López and Marta Luna-Sánchez
 Manuscript Number: EMM-2016-06345